# Deep Recurrent Optimal Stopping

**Niranjan Damera Venkata**
Digital and Transformation Organization
HP Inc., Chennai, India
`niranjan.damera.venkata@hp.com`

**Chiranjib Bhattacharyya**
Dept. of CSA and RBCCPS
Indian Institute of Science, Bangalore, India
`chiru@iisc.ac.in`

## Abstract

Deep neural networks (DNNs) have recently emerged as a powerful paradigm for solving Markovian optimal stopping problems. However, a ready extension of DNN-based methods to non-Markovian settings requires significant state and parameter space expansion, manifesting the curse of dimensionality. Further, efficient state-space transformations permitting Markovian approximations, such as those afforded by recurrent neural networks (RNNs), are either structurally infeasible or are confounded by the curse of non-Markovianity. Considering these issues, we introduce, for the first time, an optimal stopping policy gradient algorithm (OSPG) that can leverage RNNs effectively in non-Markovian settings by implicitly optimizing value functions without recursion, mitigating the curse of non-Markovianity. The OSPG algorithm is derived from an inference procedure on a novel Bayesian network representation of discrete-time non-Markovian optimal stopping trajectories and, as a consequence, yields an offline policy gradient algorithm that eliminates expensive Monte Carlo policy rollouts.

## 1 Introduction

In the classic optimal stopping setting, an agent monitors the trajectory of a stochastic process, with the opportunity to either continue observing or stop and claim a reward (or suffer a cost), which is generally a function of process history. Once the agent decides to stop, no further actions are possible, and no further reward can be claimed (or cost suffered). The agent's goal is to produce stopping decisions based on the process observations to maximize the expected reward (or minimize the expected cost). For example, whether to exercise a stock option or not at any time is an optimal stopping problem where the goal is to maximize expected exercise value, as is the decision of when to replace a degrading machine. Optimal stopping methods have a wide array of applications in finance [5, 2], operations research [17, 24, 10], disorder/change-point detection [36, 32], early classification of time-series [13, 1] among others.

In this paper, we are concerned with the *discrete-time, finite-horizon, model-free* setting, where the process evolves over discrete time-steps, and a decision to stop must occur by a given finite horizon. Further, the dynamics of process evolution are unknown. This setting is in contrast with classic approaches [8, 28, 35, 14, 36] which assume that the dynamics of process evolution are known and characterized with simplified and usually Markovian stochastic models. Note that continuous-time optimal stopping problems, including the pricing of American options, may often be reduced to the discrete-time setting by suitable discretization [18]. Solving optimal stopping problems in non-Markovian settings is challenging due to the following curses:

- **The curse of dimensionality**[3, 4, 33]: A non-Markovian process may be made Markovian by extending state space with process history. However, such an extension significantly increases the problem's dimensionality, complicating function approximation. Estimating high-dimensional value functions directly rules out conventional approaches that approximate value functions with a linear combination of basis functions [40, 23, 39, 40, 43].

37th Conference on Neural Information Processing Systems (NeurIPS 2023).

- **The curse of non-Markovianity** [41, 6]: One may mitigate the curse of dimensionality by approximating a non-Markovian process by a Markovian process with a transformed state-space. For example, RNNs can learn to summarize relevant process history into a Markovian hidden-state process efficiently. However, state-of-the-art Markovian optimal stopping methods [19, 20, 2, 16] rely on the recursive computation of value functions (see Section 2) which propagates approximation bias in non-Markovian settings, a phenomenon known as the curse of non-Markovianity [6].

**Existing deep neural network approaches:** Deep neural network approaches to optimal stopping [19, 20, 2, 16] have emerged as a powerful way of handling high-dimensional Markovian optimal stopping problems. Existing model-free methods compute value function approximations relying on a recursive temporal relationship between value functions at successive time-steps in Markovian settings embodied by the Wald-Bellman equation [36] or closely related equivalent (see Section 2).

*Backward Induction* methods [19, 20, 2, 16] start from the last time-step and recurse backward sequentially through time. Non-Markovian settings force Backward Induction methods to augment their state space with process history [2] significantly exacerbating the curse of dimensionality since elegant approaches such as RNNs, which share parameters across time-steps are not an option for backward induction methods. If the value function of time-step $j + 1$ is already optimized, it cannot change when we fit the value function at time-step $j$. Further, the inability to share parameters across time-steps means parameter space grows linearly with the time-steps processed, thereby reducing sample efficiency. A notable exception is the method of Herrera et al. [16], which uses an RNN. However, the RNN weights are random and not learnable to permit backward induction to proceed. Only the final hidden-layer weights are fit, and these are not shareable across time-steps.

*Fitted Q-Iteration* (FQI) methods [34, 16] start with bootstrapped value functions at all time-steps and recursively update these for temporal consistency with the Wald-Bellman equation. While they can use RNN-based function approximators, they suffer from the curse of non-Markovianity.

We note that a body of work uses deep neural networks to solve continuous-time optimal stopping problems by formulating them as free-boundary Partial Differential Equations (PDE). These include methods such as the deep Galerkin method [37] and the backward stochastic differential equation method [7]. We consider such methods model-based since they start with a specific PDE to be solved that assumes prior knowledge of process evolution.

**Outline of contributions:** The above context points to (1) using a suitable characterization of state space (as afforded by RNNs) to mitigate the curse of dimensionality and (2) direct policy estimation methods such as policy gradients to mitigate the curse of non-Markovianity. Our approach brings together, for the first time, probabilistic graphical models, policy gradient methods [42], and RNNs to design effective stopping policies for non-Markovian settings. policy gradient methods are notably missing from the optimal stopping literature. We make the following contributions:

- We present a reward augmented trajectory model (RATM) (see Section 3), a novel Bayes net parameterization of non-Markovian state-action-reward trajectories encountered in optimal stopping, where conditional probability distributions (CPDs) encode stochastic policy actions and reward possibilities (Section 3). In particular, this formulation allows the stochastic policy to share parameters across time-steps, allowing RNN-based CPD approximations, which mitigate the curse of dimensionality issues discussed earlier. Also, the problem of finding an optimal stopping policy reduces to a direct policy optimization procedure based on the E-M algorithm (Theorem 3.1, and Corollary 3.1.1) that leverages inference over the Bayes net. This mitigates the curse of non-Markovianity associated with recursive value function estimation. Further, modeling optimal stopping with graphical models opens new avenues. As noted by Levine [22], in the context of reinforcement learning, "The extensibility and compositionality of graphical models can likely be leveraged to produce more sophisticated reinforcement learning methods, and the framework of probabilistic inference can offer a powerful toolkit for deriving effective and convergent learning algorithms for the corresponding models"; for instance, augmentation with additional latent variables may be explored to model process disorders such as change-points.

- Inference on the RATM yields a policy gradient algorithm when we take an incremental view of the E-M algorithm (see Section 4, Theorem 4.1 and Proposition 4.1). We call this algorithm the optimal stopping policy gradient (OSPG) method, the first policy gradient algorithm, to the best of our knowledge, for computing optimal stopping policies suitable for non-Markovian settings. A key benefit of OSPG is that unlike classic policy gradients [42], it is an offline algorithm and does

not require Monte-Carlo policy rollouts. Crucially, we show (Corollary 4.1.1) that our procedure is learning a policy that implicitly optimizes value functions at every time-step without requiring recursive value function computation, thereby mitigating the curse of non-Markovianity.

- We introduce a new loss function based on OSPG that is well-suited for learning RNN-based stopping policies with mini-batch stochastic gradient descent (Algorithm 1). This makes OSPG (unlike E-M) practical for large datasets while leveraging powerful off-the-shelf optimizers such as Adam. The OSPG loss considers an entire process trajectory and does not decompose across time-steps, allowing policy decisions across a trajectory to be evaluated jointly rather than independently.

## 2 Discrete-time finite-horizon optimal stopping

We represent the stochastic process monitored by $d$-dimensional random fully observable environment state variables $\{S_j\}$. Process history until and including time-step $j$ is represented by the $d \times j$ random matrix $\mathbf{S}_j := [S_0, S_1, \cdots S_j]$. Associated with state history is a reward process $\{R_j\}$ with $R_j = g_j(\mathbf{S}_j)$, where $g_j : \mathbb{R}^{d \times j} \mapsto \mathbb{R}^+$ is a reward function that maps observation history to a reward for stopping at time-step $j$. Uppercase letters denote random variables, while lowercase letters represent their realizations. Bold font is used to indicate histories. Thus $\mathbf{s}_j$ is an observed partial trajectory from $\mathbf{S}_j$. Also, in case we need to refer to a specific trajectory, we use an extra index $i$, so for example, $r_{ij}$ refers to the observed reward for stopping at time-step $j$ when monitoring the $i^{th}$ process trajectory.

**Definition 2.1** (stopping policy). A stopping policy is defined as a sequence of functions $\varphi = (\varphi_1, \varphi_2, \cdots \varphi_j, \cdots)$, where $\varphi_j : \mathbb{R}^{d \times j} \mapsto \{0, 1\}$ maps the history of observations at a time-step $j$ to a decision of whether to stop ($\varphi_j(\mathbf{s}_j) = 1$) or not ($\varphi_j(\mathbf{s}_j) = 0$).

**Definition 2.2** (stochastic stopping policy). A stochastic stopping policy is defined as a sequence of functions $\boldsymbol{\phi} = (\phi_1, \phi_2, \cdots \phi_j, \cdots)$, where $\phi_j : \mathbb{R}^{d \times j} \mapsto [0, 1]$ maps the history of observations at a time-step $j$ to a probability of stopping at time $j$.

Note that a stochastic policy may be converted to a deterministic stopping policy to make stopping decisions by augmenting the state space with i.i.d. virtual observations from a random variable $U$ uniformly distributed in $[0, 1]$ Thus, the equivalent deterministic policy is $\varphi_j(\mathbf{s}_j) := \mathbb{I}(\phi_j(\mathbf{s}_j) \geq u_j)$.

This paper considers the finite horizon setting where $j \leq H \in \mathbb{N}$, so a decision to stop must be made at or before a finite horizon $H$. Thus we define $\varphi_H(\mathbf{S}_H) := 1$. The decision to stop is based on a stochastic process, making the stopping time a random variable.

**Definition 2.3** (policy stopping time [31]). Stopping occurs when the policy first triggers a stop action. This occurs at random time $\tau := \min\{0 \leq j \leq H : \varphi_j(\mathbf{S}_j) = 1\}$, called the stopping time[1].

With this background, we can formally define the optimal stopping problem:

**Definition 2.4** (optimal stopping problem [36]). Solve for the optimal stopping time $\tau^*$ (if one exists) for which $\mathbb{E}[R_{\tau^*}] = \sup_\tau \mathbb{E}[R_\tau]$ where $R_j = g_j(\mathbf{S}_j)$ is a reward process, $\tau$ is a stopping time random variable, and $\{\mathbf{S}_j\}$ is a stochastic process.

Following Shiryaev and Peskir [36, 28], in the finite-horizon, discrete-time setting, the existence of a finite optimal stopping-time is guaranteed when $\mathbb{E}\left[\sup_{j \leq H} |R_j|\right] < \infty$. The optimal stopping time can then be determined by the method of *backward induction* which recursively constructs the *Snell-envelope*[2] comprising random variables $\{U_j\}$ given by:

$$U_H := R_H \ , \ U_j = \max\{R_j, \mathbb{E}[U_{j+1}|\mathbf{S}_j]\} \ , \ j = H - 1, \cdots 0 \qquad \text{(Snell-envelope)}$$

The optimal stopping time is given by $\tau^* = \min\{0 \leq j \leq H : U_j = R_j\}$. From a practical perspective, the expectations in the recursion are hard to estimate since they depend on process history, manifesting the curse of dimensionality. The assumption of Markovian state evolution, i.e. $\mathbb{P}(S_{j+1}|\mathbf{S}_j) = \mathbb{P}(S_{j+1}|S_j)$, and a reward process expressable as a function $G_j$ of the current state,

---

[1]This is equivalent (see [15]) to the classic definition [36] of a stopping time w.r.t a filtration $\{\mathcal{F}_j\}$ as a RV such that $\{\tau = j\} \in \mathcal{F}_j \forall j$. Since $\{\mathcal{F}_j\}$ is the minimal filtration generated by $\{\mathbf{S}_j\}$ in our setting, we can determine if $\tau = j$ or not by considering $\mathbf{s}_j$. We use lowercase to represent $\tau$ for consistency with literature.

[2]The Snell-envelope is also the smallest super-martingale that dominates/envelopes the reward process [36].

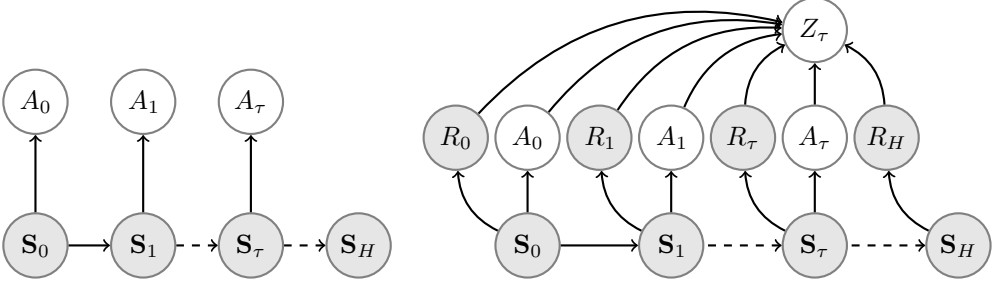

(a) state-action trajectory model

(b) reward augmented Bayesian network model

Figure 1: Bayesian networks representing (a) state-history ($\mathbf{S}_j$) and action ($A_j$) trajectories and (b) corresponding reward augmented trajectory model for optimal stopping.

i.e. $R_j = G_j(S_j)$, directly leads to the much more tractable *Wald-Bellman equation* [28, 35, 36]:

$$V_H(s) := G_H(s) \ , \ V_j(s) = \max\{G_j(s), T_j V_{j+1}(s)\}, j < H \qquad \text{(Wald-Bellman equation)}$$

The *value functions* $V_j(s)$ give the maximum expected reward obtainable from time $j$ having observed state $s$. Thus $V_j(s) = \sup_{\tau \geq j} \mathbb{E}_s[R_\tau] \ \forall j$. The operator $T_j$ is defined by $T_j F(s) := \mathbb{E}_{j,s}[F(S_{j+1})]$ for any function $F$ of the state. The Snell-envelope reduces to $U_j = V_j(S_j)$ in this case. We define $K_j(s) := T_j V_{j+1}(s)$ as the *continuation value-function*, which is the maximum expected reward one can obtain by choosing to continue at step $j$ having observed state $s$. The optimal stopping policy is given by:

$$\varphi_H^*(\mathbf{S}_j) := 1 \ , \ \varphi_j^*(\mathbf{S}_j) = \mathbb{I}(G_j(S_j) \geq K_j(S_j)), \ j = 0, 1, \cdots H - 1 \qquad (1)$$

The value functions are, therefore, related to the optimal policy as follows:

$$V_j(S_j) = \varphi_j^*(S_j) G_j(S_j) + (1 - \varphi_j^*(S_j)) K_j(S_j) \qquad (2)$$

Thus, if we can estimate the continuation value functions $K_j$, we can trigger optimal stopping decisions using equation (1). Note that this requires knowledge of reward function $G_j$ at inference time. Backward induction approaches typically, first estimate $K_{H-1}$ to approximate $K_{H-1}(S_{H-1}) \approx G_H(S_H)$ and then use either the Wald-Bellman equation [40, 19] or equations (1) and (2) [23, 20, 16] to produce approximation targets $V_{H-1}(S_{H-1})$ which are used as targets to fit $K_{H-2}(S_{H-2}) \approx V_{H-1}(S_{H-1})$, and the process continues backwards recursively. As a variation on this theme, Becker et al. [2] first set $K_{H-1}(S_{H-1}) = G_H(S_H)$ and then estimate $\varphi_j^*(s)$ in equation (2) by maximizing the RHS of equation (2) with $j = H - 1$ and $\varphi_{H-1}$ replacing $\varphi_{H-1}^*$. Then equation (2) is used to compute $V_{H-1}(S_{H-1})$ which is in turn used to set $K_{H-2}(S_{H-2}) = V_{H-1}(S_{H-1})$ and the process proceeds recursively. Fitted Q-iteration methods [39, 40, 43] use bootstrapped continuation value functions $K_j$ to produce $V_j(S_j)$ at all time-steps using the Wald-Bellman equation. Then, updated continuation value-functions $K_j$ are fit simultaneously across all time-steps to approximate $V_{j+1}(S_{j+1})$, and the process continues recursively until convergence. If Markovian assumptions do not hold, states $S_j$ need to be replaced with $\mathbf{S}_j$ exploding the state space and significantly complicating function approximation (the curse of dimensionality). The recursive procedures discussed above propagate bias across time-steps when using Markovian approximations, since bias in value estimation at one time-step infects the preceding time-step (the curse of non-Markovianity).

## 3 A Bayesian network view of optimal stopping

This section presents a new Bayesian network view of optimal stopping. While we draw inspiration from a corresponding view of general reinforcement learning (RL) [22], the resulting modeling choices needed to capture the structure of optimal stopping make the approaches distinct.

**State-action trajectory model:** Figure 1(a) shows a Bayes net model of state-action trajectories in optimal stopping. An action $A_j$ at any time-step $j$ depends on the state history $\mathbf{S}_j$ up to that time-step, permitting general non-Markovian stopping policies. Further, actions are restricted to 0 (continue) or 1 (stop), with the stop action terminating the action trajectory. Therefore, a key feature of optimal stopping problems is that, unlike general reinforcement learning, actions do not impact the probability distribution of the following states over the control horizon, conditioned on state history. We do not model the stopping time $\tau$ explicitly. Instead, it is a consequence of stochastic policy actions over

the horizon. The action trajectory stops at a random stopping time $\tau = \min\{0 \leq j \leq H : A_j = 1\}$. Therefore, we have the special structure $A_\tau = 1$ and $A_j = 0, \forall j < \tau$. Note that Figure 1(a) represents a family of stopping trajectories $\mathbf{A}_\tau$, each member stopping at a specific value of $\tau$. We denote a specific stopping trajectory by $\mathbf{A}_j$. Consider stochastic policies, given by the functions $\phi_j(\mathbf{S}_j) := \mathbb{P}(A_j = 1|\mathbf{S}_j)$. We define $\psi_j(\mathbf{S}_j) := \mathbb{P}(\tau = j|\mathbf{S}_j)$ to represent the stopping-time distribution. We have the following key relationship between action trajectories and the stopping-time random variable encapsulated in the following lemma, adapted from [14], proved in Appendix A.

**Lemma 3.1** (trajectory reparameterization lemma)**.**

$$\mathbb{P}(\tau = j|\mathbf{S}_H) = \psi_j(\mathbf{S}_j) = \mathbb{P}(\mathbf{A}_j|\mathbf{S}_j) = \begin{cases} \phi_0(\mathbf{S}_0) & \text{if } j = 0 \\ \phi_j(\mathbf{S}_j) \prod_{n=0}^{j-1}(1 - \phi_n(\mathbf{S}_n)), & \text{if } 0 < j < H \\ \prod_{n=0}^{H-1}(1 - \phi_n(\mathbf{S}_n)), & \text{if } j = H \end{cases}$$

**Reward augmented trajectory model:** Figure 1b shows an augmentation of the basic state action trajectory model to bring the notion of stopping rewards into the model. The variables $R_j$ represent the reward process, which are functions of state history: recall $R_j = g_j(\mathbf{S}_j)$. We introduce $Z_\tau$ as a family of binary random variables representing if a reward is obtained ($Z_\tau = 1$) or not ($Z_\tau = 0$) when we stop at various values of $\tau$. Thus stopping at $j$ no longer guarantees reward $R_j$. The idea is to use random variables $Z_\tau$ to transform absolute rewards which are not readily interpreted as probabilities into probabilities (in this case, probabilities of obtaining a reward at any time-step) that may be represented in the Bayes net. We parameterize the conditional probability distribution of $Z_\tau$ with Bernoulli distributions defined as follows:

$$\mathbb{P}(Z_\tau = 1|\mathbf{R}_H, \mathbf{A}_\tau) = \tilde{R}_\tau := \tilde{r}(\tau, \mathbf{R}_H) \quad \mathbb{P}(Z_\tau = 0|\mathbf{R}_H, \mathbf{A}_\tau) = 1 - \tilde{R}_\tau \tag{3}$$

where $\tilde{r} : \{0, 1, \cdots H\} \times (\mathbb{R}^+)^H \mapsto [0, 1]$ is a function that transforms real-valued stopping rewards into probabilities that encode the notion of the *relative* possibility of obtaining rewards over a trajectory by stopping at a particular time-step of the trajectory. We may write the joint distribution of a trajectory stopping at time $\tau$ as:

$$\mathbb{P}(\mathbf{A}_\tau, Z_\tau, \mathbf{R}_H, \mathbf{S}_H) = \mathbb{P}(\mathbf{S}_0)\mathbb{P}(\mathbf{R}_0|\mathbf{S}_0)\underbrace{\prod_{j=1}^{H}\mathbb{P}(\mathbf{S}_j|\mathbf{S}_{j-1})\mathbb{P}(\mathbf{R}_j|\mathbf{S}_j)}_{\mathbb{P}(\mathbf{S}_H)\mathbb{P}(\mathbf{R}_H|\mathbf{S}_H)}\underbrace{\prod_{n=0}^{\tau}\mathbb{P}(A_n|\mathbf{S}_n)}_{\mathbb{P}(\mathbf{A}_\tau|\mathbf{S}_\tau)}\mathbb{P}(Z_\tau|\mathbf{A}_\tau, \mathbf{R}_H)$$

$$= \mathbb{P}(\mathbf{S}_H)\mathbb{P}(\mathbf{R}_H|\mathbf{S}_H)\mathbb{P}(\tau|\mathbf{S}_H)\mathbb{P}(Z_\tau|\mathbf{A}_\tau, \mathbf{R}_H) \tag{4}$$

We have used the reparameterization lemma 3.1 to replace the probability distribution over actions with the corresponding induced probability distribution over stopping times. Conditioning on $\mathbf{R}_H$ and $\mathbf{S}_H$ we have:

$$\mathbb{P}(\mathbf{A}_\tau, Z_\tau|\mathbf{R}_H, \mathbf{S}_H) = \mathbb{P}(\tau|\mathbf{S}_H)\mathbb{P}(Z_\tau|\mathbf{A}_\tau, \mathbf{R}_H) \tag{5}$$

**Optimal policy estimation:** We are particularly interested in the conditional probability of getting a reward when we stop. We can obtain this by using Bayes net inference to obtain equation (5) and then marginalizing over the family of stopping trajectories $\mathbf{A}_\tau$. Since there is a one-to-one correspondence between $\mathbf{A}_\tau$ and stopping time $\tau$, we just need to sum over all possible stopping times. Thus, if we define $Z := Z_0 \oplus Z_1 \oplus \cdots \oplus Z_H$, where $\oplus$ is the XOR operator[3], we have, from equation (5):

$$\mathbb{P}(Z = 1|\mathbf{R}_H, \mathbf{S}_H) = \sum_{j=0}^{H}\mathbb{P}(\tau = j|\mathbf{S}_H)\mathbb{P}(Z_j = 1|\mathbf{A}_j, \mathbf{R}_H) = \sum_{j=0}^{H}\psi_j(\mathbf{S}_j)\tilde{R}_j \tag{6}$$

Ideally, we would want our policy to maximize this conditional probability. To this end, we consider parameterized policy functions $\phi_j^{\boldsymbol{\theta}}$ and induced stopping time distributions $\psi_j^{\boldsymbol{\theta}}$ and define the conditional likelihood function over a sample trajectory as:

$$l(\boldsymbol{\theta}|\mathbf{s}_H, \mathbf{r}_H) = \mathbb{P}(Z = 1|\mathbf{s}_H, r_H; \boldsymbol{\theta}) = \sum_{j=0}^{H}\psi_j^{\boldsymbol{\theta}}(\mathbf{s}_j)\tilde{r}_j \tag{7}$$

---

[3]We use the XOR operator since exactly one of the $\{Z_j\}$ can be equal to one. The reward for a trajectory can only be claimed by a stop action at a single time-step.

However, maximizing this likelihood over samples while allowing optimal stopping decisions on any trajectory ignores the relative magnitude of rewards between different trajectories. Thus, we seek to weight the likelihood of each trajectory. Therefore, considering a dataset $\mathcal{D}$ of environment trajectories and using index $i$ for the $i^{th}$ trajectory in the sample, we have:

$$l_{\text{wml}}(\boldsymbol{\theta}) = \prod_{i=1}^{N} l(\boldsymbol{\theta} \,|\, \mathbf{s}_{i,H}, \mathbf{r}_{i,H})^{\tilde{w}_i} \quad \text{with} \quad \sum_{i=1}^{N} \tilde{w}_i = 1 \tag{8}$$

where $\tilde{w}_i$ is the relative weight for the $i^{th}$ sample. Taking logarithms, we have the following objective to maximize:

$$J_{\text{WML}}(\boldsymbol{\theta}) = \sum_{i=1}^{N} \tilde{w}_i \log \sum_{j=0}^{H} \psi_j^{\boldsymbol{\theta}}(\mathbf{s}_{ij}) \tilde{r}_{ij} \tag{9}$$

We may use the following Expectation-Maximization (E-M) procedure to maximize this objective:

**Theorem 3.1** (E-M procedure for optimal stopping policy). *The following iterative procedure converges to a local maximum of $J_{WML}(\boldsymbol{\theta})$ with monotonic increase in the objective with iteration index $(k)$, starting with an initial value $\boldsymbol{\theta} = \boldsymbol{\theta}^{(0)}$:*

$$q_{ij}^{(k)} = \frac{\psi_j^{\boldsymbol{\theta}^{(k)}}(\mathbf{s}_{ij}) \tilde{r}_{ij}}{\sum_{j=0}^{H} \psi_j^{\boldsymbol{\theta}^{(k)}}(\mathbf{s}_{ij}) \tilde{r}_{ij}} \quad 0 \leq j \leq H, \ 1 \leq i \leq N \tag{E-step}$$

$$J_M^{(k)}(\boldsymbol{\theta}) = \sum_{i=1}^{N} \sum_{j=0}^{H} \tilde{w}_i^{(k)} q_{ij}^{(k)} \log \psi_j^{\boldsymbol{\theta}}(\mathbf{s}_{ij}) \,, \ \boldsymbol{\theta}^{(k+1)} = \arg\max_{\boldsymbol{\theta}} J_M^{(k)}(\boldsymbol{\theta}) \tag{M-step}$$

where $\tilde{w}_i^{(k)} = \tilde{w}_i, \forall k$. The proof follows from an application of Jensen's inequality and is given in Appendix A. Dayan and Hinton pioneered using E-M methods for RL policy search [12].

**Specification of $\{\tilde{r}_{ij}\}$ and $\tilde{\mathbf{w}}$:** $\tilde{r}_{ij}$ is a specification of the relative *intra-trajectory* probability of obtaining a reward when stopping at time-step $j$ of the $i^{th}$ trajectory. A natural choice (and one that we adopt in this paper) is to set $\tilde{r}_{ij} := \frac{r_{ij}}{\sum_{j=0}^{H} r_{ij}}$, where $r_{ij}$ is a realization of $R_j = g_j(\mathbf{S}_j)$ corresponding to the $i^{th}$ trajectory. This formulation encourages stopping at time-steps of the trajectory, proportional to achievable reward. Alternatively, formulations such as $\tilde{r}_{ij} := \frac{r_{ij}}{\max_k r_{ik}}$ or $\tilde{r}_{ij} := \frac{\exp(r_{ij})}{\sum_{k=0}^{H} \exp(r_{ik})}$ could be used to express different stopping preferences within a trajectory. The weights $\tilde{\mathbf{w}}$, on the other hand, express relative *inter-trajectory* importance. One plausible choice would be to set $\tilde{w}_i = \tilde{r}_i = \frac{\sum_{j=0}^{H} r_{ij}}{\sum_{i=1}^{N} \sum_{j=0}^{H} r_{ij}}$, the relative reward available for extraction from a trajectory. Similarly, one could weight trajectories by the maximum reward achievable: $\tilde{w}_i = \tilde{r}_i = \frac{\max_j r_{ij}}{\sum_i \max_j r_{ij}}$. We may also consider reweighting schemes that vary weights over rounds of the E-M algorithm. In the RL literature, Reward-weighted regression [30] takes such an approach. Reward-weighted regression typically conducts rollouts of the current policy and weights each resulting sample trajectory with the relative reward achieved by the current policy on that trajectory. However, the trajectory reward is unknown since we do not perform policy rollouts. Instead, we weight each trajectory offline by its relative expected reward under the current policy. Specifically, we compute weights at each round according to a W-step:

$$\tilde{w}_i^{(k)} = \frac{\sum_{j=0}^{H} \psi_j^{\boldsymbol{\theta}^{(k)}}(\mathbf{s}_{ij}) r_{ij}}{\sum_{i=1}^{N} \sum_{j=0}^{H} \psi_j^{\boldsymbol{\theta}^{(k)}}(\mathbf{s}_{ij}) r_{ij}} \quad 1 \leq i \leq N \tag{W-step}$$

This weighting is not arbitrary as it leads to a surprising equivalence with a policy gradient method, established in the next section. Further, we have the following Corollary (proof in Appendix A):

**Corollary 3.1.1** (Reweighted E-M). *Let $\tilde{r}_i = \frac{\sum_{j=0}^{H} r_{ij}}{\sum_{i=1}^{N} \sum_{j=0}^{H} r_{ij}}$. Using the reweighting scheme of the W-step in the E-M procedure of Theorem 3.1 achieves a local maximum of:*

$$J_{WML}(\tilde{\mathbf{w}}, \boldsymbol{\theta}) = \sum_{i=1}^{N} \tilde{w}_i \log \sum_{j=0}^{H} \psi_j^{\boldsymbol{\theta}}(\mathbf{s}_{ij}) \tilde{r}_{ij} - \sum_{i=1}^{N} \tilde{w}_i \log \frac{\tilde{w}_i}{\tilde{r}_i} \tag{10}$$

*with monotonic increase in the objective, starting with an initial value $\boldsymbol{\theta} = \boldsymbol{\theta}^{(0)}$. The second term is the KL-divergence between distributions $\tilde{\mathbf{w}}$ and $\tilde{\mathbf{r}}$.*

# 4 Optimal stopping policy gradients (OSPG)

In this section we show that an incremental version of the E-M algorithm described in the previous section, with a single gradient step replacing a full M-step, is equivalent to a policy gradient method that maximizes expected reward. We call this the optimal stopping policy gradient (OSPG) method.

Consider a direct maximization of the optimal stopping objective $J_{OS}(\boldsymbol{\theta}) = \mathbb{E}[R_\tau]$ from definition 2.4 where the expectation is over state-action trajectories. We can use our Bayes net model of state-action trajectories without reward augmentation shown in Figure 1a to obtain:

$$\mathbb{P}(\mathbf{S}_H, \mathbf{A}_\tau | \boldsymbol{\theta}) = \mathbb{P}(\mathbf{S}_0) \prod_{j=1}^{H} \mathbb{P}(\mathbf{S}_j | \mathbf{S}_{j-1}) \prod_{n=0}^{\tau} \mathbb{P}(A_n | \mathbf{s}_n, \boldsymbol{\theta}) = \mathbb{P}(\mathbf{S}_H)\mathbb{P}(\tau | \mathbf{S}_H, \boldsymbol{\theta}) \quad (11)$$

where we have used the trajectory reparameterization lemma to reparameterize in terms of stopping times. Therefore, we may write:

$$J_{OS}(\boldsymbol{\theta}) = \mathbb{E}_{\mathbf{s}_H \sim \mathbb{P}(\mathbf{s}_H)} \left[ \mathbb{E}_{\tau \sim \mathbb{P}(\tau | \mathbf{s}_H, \boldsymbol{\theta})} [R_\tau] \right] = \mathbb{E}_{\mathbf{s}_H \sim \mathbb{P}(\mathbf{s}_H)} \left[ \sum_{j=0}^{H} \psi_j^{\boldsymbol{\theta}}(\mathbf{s}_j) r_j \right] \quad (12)$$

We leverage methods used to establish the policy gradient theorem [42, 38] to derive a convenient form for the gradient of the optimal stopping objective w.r.t policy parameters $\boldsymbol{\theta}$.

**Theorem 4.1** (Optimal stopping policy gradient theorem). *For any differentiable stopping policy $\phi^{\boldsymbol{\theta}}$, The gradient of the objective $J_{OS}(\boldsymbol{\theta}) = \mathbb{E}[R_\tau]$ w.r.t. policy parameters $\boldsymbol{\theta}$, where $\{R_j\}$ is a risk process and $\tau$ is a stopping-time random variable, is given by:*

$$\nabla_{\boldsymbol{\theta}} J_{OS}(\boldsymbol{\theta}) = \mathbb{E}_{\mathbf{s}_H \sim \mathbb{P}(\mathbf{s}_H)} \left[ \sum_{j=0}^{H} r_j \psi_j^{\boldsymbol{\theta}}(\mathbf{s}_j) \nabla_{\boldsymbol{\theta}} \log \psi_j^{\boldsymbol{\theta}}(\mathbf{s}_j) \right]$$

*where the $\psi_j^{\boldsymbol{\theta}}(\mathbf{s}_j)$ are functions of $\phi^{\boldsymbol{\theta}}$ obtained by Lemma 3.1.*

The proof is an application of the log-derivative trick [42] and is given in Appendix A. Note that optimal stopping policy gradients are estimated using offline state trajectories only. Unlike the general RL case, we never need to do policy rollouts to sample trajectories. Instead, we explicitly calculate the probability distribution over all possible stopping actions (or stopping times) and use this to average over all possible stop-continue decisions. Also, an update to the policy parameters can be done without fresh data collection since we can correctly (due to explicit Bayes net modeling) model the impact of the updated policy on the loss using the already sampled environment trajectories. Appendix B describes specialization to settings with costs instead of rewards.

We now establish a surprising equivalence between the reweighted E-M algorithm of Corollary 3.1.1 and the policy gradient method derived in this section when an incremental E-M solution approach [27] is used. Although the objective functions $J_{WML}(\tilde{\mathbf{w}}, \boldsymbol{\theta})$ and $J_{OS}(\boldsymbol{\theta})$ are different in general, the same procedure maximizes both.

**Proposition 4.1** (Incremental E-M as a policy gradient method). If a partial M-step is used in the reweighted E-M approach of Corollary 3.1.1, comprising of a single gradient-step, then the resulting incremental E-M procedure is equivalent to the optimal stopping policy gradient method of Theorem 4.1, each converging to a local maximum of both $J_{WML}(\tilde{\mathbf{w}}, \boldsymbol{\theta})$ and $J_{OS}(\boldsymbol{\theta})$.

The proof is given in Appendix A. Thus, we may view the incremental E-M approach that optimizes a weighted maximum likelihood objective as a generalization of the policy gradient method that seeks to optimize the expected reward objective. The two are equivalent for a particular weighting scheme and Bayes net specification (as in Corollary 3.1.1). Further, the OSPG gradient has the following insightful form, obtained by applying Lemma 3.1 to the M-step objective of Theorem 3.1.

**Corollary 4.1.1** (Value form of the OSPG gradient). *Let $v_{ij} := \psi_j^{\boldsymbol{\theta}}(\mathbf{s}_{ij}) r_{ij}$, $k_{ij} := \left[ \sum_{n=j+1}^{H} v_{in} \right]$. We may rewrite the policy gradient of Theorem 4.1 as:*

$$\nabla_{\boldsymbol{\theta}} J_{OS}(\boldsymbol{\theta}) = \frac{1}{N} \sum_{i=1}^{N} \sum_{j=0}^{H} \left[ \frac{v_{ij}(1 - \phi_j^{\boldsymbol{\theta}}(\mathbf{s}_{ij})) - k_{ij} \phi_j^{\boldsymbol{\theta}}(\mathbf{s}_{ij})}{\phi_j^{\boldsymbol{\theta}}(\mathbf{s}_{ij})(1 - \phi_j^{\boldsymbol{\theta}}(\mathbf{s}_{ij}))} \right] \nabla_{\boldsymbol{\theta}} \phi_j^{\boldsymbol{\theta}}(\mathbf{s}_{ij}) \quad (13)$$

The proof is given in Appendix A. $v_{ij}$ is an empirical estimate of immediate reward, while $k_{ij}$ is an empirical estimate of continuation value under the current policy. The term in the bracket is the the derivative of the objective w.r.t. the policy output, evaluated at $\mathbf{s}_{ij}$. This is positive, and hence calls for increasing stopping probability, if $\frac{v_{ij}}{k_{ij}} > \frac{\phi_j^{\boldsymbol{\theta}}(\mathbf{s}_{ij})}{1 - \phi_j^{\boldsymbol{\theta}}(\mathbf{s}_{ij})}$, i.e. when the ratio of immediate to continuation value exceeds the current odds of stopping. Of course, the policy is updated considering expected immediate reward and continuation values at all time-steps. The updated policy produces new estimates of immediate reward and continuation value from which a new OSPG gradient is computed, and process proceeds until convergence to an optimal policy. Thus, this process implicitly uses value functions at all time-steps to update the policy which implicitly further improves value functions. Since there is no recursion across time-steps OSPG is able to mitigate the curse of non-Markovianity.

**Temporal loss functions for learning RNN stopping policies:** We may directly adopt the M-step objective of Theorem 3.1 as a loss function for learning an RNN-based policy with mini-batch stochastic gradient-descent (SGD), effective in non-Markovian settings. Algorithm 1 shows the pseudocode for computing our OSPG loss. This loss is ideally suited for learning stopping policies with RNNs since it takes as input stopping probabilities at every time-step $\phi_j^{\boldsymbol{\theta}}(\mathbf{s}_{ij})$ that are each a function of process observation up to that point. An RNN performs this mapping naturally since it maps input history to a hidden state at each time-step with recurrent computation. So, for example, return_sequences=True, coupled with a TimeDistributed Dense layer in Keras, returns a complete trajectory of RNN predictions for input into the loss. Note also that this is a temporal loss that considers the entire process trajectory and does not decompose across time-steps.

---

**Algorithm 1** Pseudocode for mini-batch computation of our temporal OSPG loss

---

   **Input:** $\mathbf{R} := [r_{ij}]$, $\Phi := \left[\phi_j^{\boldsymbol{\theta}}(\mathbf{s}_{ij})\right]$ $\{1 \leq i \leq N_b, 0 \leq j \leq H, N_b \text{ is batch size}\}$

   $\Psi_{:,0} = \Phi_{:,0}$, $\Psi_{:,H} = \exp\left(\sum_{j=0}^{H-1} \log(1 - \Phi_{:,j})\right)$ $\{$numerically stable computation of $\psi_j^{\boldsymbol{\theta}}(\mathbf{s}_{ij})\}$

   $\Psi_{:,1:H-1} = \exp\left(\log(1 - \Phi_{:,H-2})\mathbf{U} + \log \Phi_{:,1:H-1}\right)$ $\{\mathbf{U}$ is unit upper triangular, CUMSUM$\}$

   $\mathbf{V} = \text{stop-gradient}(\mathbf{R} \odot \Psi)$ $\{$held constant, so no gradient. $\odot$ denotes Schur product$\}$

   **Output:** $J = -\frac{1}{N_b}\mathbf{1}^T[\mathbf{V} \odot \log \Psi]\mathbf{1}$

---

## 5 Experiments

We compare our OSPG algorithm using deep neural networks (DNN-OSPG) and recurrent neural networks (RNN-OSPG) against the following model-free discrete-time optimal stopping approaches.

**Backward Induction**: Deep Optimal Stopping (DOS) [2] including an extension of state space with process history (DOS-ES) [2] and Randomized Recurrent Least Square Monte Carlo (RRLSM) [16].

**Fitted Q-iteration**: While fitted Q-Iteration (FQI) approaches in the optimal stopping literature rely on linear approximation with chosen basis functions [39, 40] break down in high-dimensional settings, they may be readily extended to leverage deep neural network (DNN-FQI) and recurrent neural network (RNN-FQI) function approximators. Appendix C provides details of our corresponding DNN-FQI and RNN-FQI baselines.

We conduct experiments with both Markovian and non-Markovian datasets. Markovian data include the pricing of high-dimensional Bermudan max-call options and American geometric-call options (relegated to Appendix D due to space). Non-Markovian data includes stopping a fractional Brownian motion and the early stopping of a sequential multi-class classifier. Except for the American option pricing experiment, all DNN function approximators use two hidden layers of 20 hidden units each, while all RNN function approximators leverage an RNN/GRU with a single hidden layer of 20 units. Backward induction methods (DOS, RRLSM) require independent networks at each time-step, while other approaches share network parameters across time-steps. Details of hyper-parameter settings, model size, and compute times are provided in Appendix D.

**Exercise of a Bermudan max-call option:** In this Markovian setting, we seek optimal strategies to maximize the returns of a Bermudan max-call option, a scenario described in [2]. Bermudan options may be exercised only at fixed times, unlike American options which may be exercised at any time prior to option expiration. The payoff of these options depends on the maximum of $d$ underlying assets with multi-dimensional Black-Scholes dynamics and the strike price $K$. The dynamics and

Table 1: Bermudan max-call option pricing: Results

| $d$ | $p_0$ | average return (standard deviation) | | | | | |
|---|---|---|---|---|---|---|---|
| | | DOS | DNN-FQI | DNN-OSPG | RNN-FQI | RNN-OSPG | RRLSM |
| 20 | 90 | 37.08 (0.09) | 36.98 (0.29) | 37.20 (0.08) | 36.98 (0.32) | **37.21** (0.07) | 35.99 (0.05) |
| 20 | 100 | 50.86 (0.08) | 50.92 (0.05) | **51.02** (0.08) | 50.76 (0.22) | 50.99 (0.09) | 49.68 (0.06) |
| 20 | 110 | 64.73 (0.11) | 64.75 (0.22) | **64.91** (0.10) | 64.67 (0.17) | 64.84 (0.10) | 63.42 (0.06) |
| 50 | 90 | 53.30 (0.08) | 53.38 (0.12) | **53.46** (0.08) | 53.43 (0.12) | 53.36 (0.11) | 51.87 (0.09) |
| 50 | 100 | 68.87 (0.12) | 68.91 (0.17) | **69.06** (0.11) | 68.98 (0.11) | 68.93 (0.07) | 67.37 (0.09) |
| 50 | 110 | 84.50 (0.13) | 84.45 (0.28) | **84.65** (0.11) | 84.62 (0.19) | 84.52 (0.16) | 82.85 (0.10) |
| 100 | 90 | 65.83 (0.10) | 65.94 (0.13) | **66.02** (0.14) | 65.78 (0.15) | 65.80 (0.16) | 64.56 (0.08) |
| 100 | 100 | 82.79 (0.16) | 82.88 (0.17) | **82.96** (0.13) | 82.82 (0.14) | 82.74 (0.17) | 81.41 (0.08) |
| 100 | 110 | 99.73 (0.16) | 99.86 (0.12) | **99.93** (0.18) | 99.74 (0.20) | 99.70 (0.14) | 98.28 (0.11) |
| 200 | 90 | 78.23 (0.15) | 78.40 (0.16) | **78.43** (0.12) | 78.11 (0.16) | 78.20 (0.09) | 77.28 (0.07) |
| 200 | 100 | 96.57 (0.16) | 96.68 (0.13) | **96.79** (0.08) | 96.21 (0.28) | 96.44 (0.13) | 95.53 (0.08) |
| 200 | 110 | 114.87 (0.10) | 115.03 (0.25) | **115.10** (0.07) | 114.55 (0.27) | 114.74 (0.23) | 113.78 (0.08) |

payoff (reward) are given by:

$$S_t^m = s_0^m \exp([r - \delta_m - \sigma_m^2/2]t + \sigma W_t^m, \quad R_t = e^{rt} \left( \max_{1 \leq m \leq d} S_t^m - K \right)^+ \quad (14)$$

for $m = 1, 2, \cdots d$. $r \in \mathbb{R}$ is the risk-free rate, $s_0^m \in (0, \infty)$ represents the initial price, $\delta_m \in [0, \infty)$ is the dividend yield, $\sigma_m \in (0, \infty)$ the volatility and $W$ is a $d$-dimensional Brownian motion, with zero instantaneous correlation between its $d$ components. The reward for exercise at time $t$ is given by $R_t$. We discretize $t$ into $H + 1$ possible exercise opportunities, using times $t_j = jT/H$ for $j = 0, 1, \cdots H$. $T$ is the option expiration duration in years. This procedure translates the problem into our discrete-time finite horizon optimal stopping setting: $\sup_{0 \leq \tau \leq H} \mathbb{E}[R_\tau]$.

Mirroring Becker et. al. [2] we set, $K = 100$, $r = 0.05$, $\sigma_m = 0.2$, $\delta_m = 0.1$, $T = 3$, $H = 9$. We generate a dataset of 40,000 sample trajectories of the underlying assets, generating different datasets for various values of $d$ and $s_0 = s_0^m, \forall m$. We train models on ten random 50% train-test splits, holding 20% of training data as a validation dataset. Means and standard deviations of the achieved exercise values by the various methods are reported in Table 1. DNN-OSPG consistently performs best, while the other Markovian DNN methods, DOS and DNN-FQI, perform well in this setting. Among the non-Markovian RNN-based methods, RLLSM fares the worst due to increased noise from prior time-steps. RNN-OSPG and RNN-FQI are still relatively competent since they can learn to ignore information from prior time-steps.

**Stopping a non-Markovian fractional Brownian motion:** We mimic the scenario considered in [2]. A fractional Brownian motion [25], $(W_t^h)_{t \geq 0}$ is a centered-Gaussian process with co-variance structure determined by its Hurst parameter $h \in (0, 1]$ as $\mathbb{E}[W_t^h W_s^h] = 0.5(t^{2h} + s^{2h} - |t-s|^{2h})$. The increments of $W^h$ are positively correlated for $h \in (0.5, 1]$ and negatively correlated for $h \in (0, 0.5)$ and uncorrelated (reducing to a standard Brownian motion) for $h = 0.5$. We discretize $t$ in the range $[0, 1]$ into time-steps by $t_j = j/H$ for $j = 0, 1, \cdots H$, and seek to solve the optimal stopping problem: $\sup_{0 \leq \tau \leq H} \mathbb{E}[W_\tau^h]$. Thus, the reward function is the identity mapping in this case. For each distinct setting of the Hurst parameter $h$, we generate 40,000 trajectories and train models on ten random 50% train-test splits, holding 20% of training data as a validation dataset.

Figure 2 plots average returns achieved by competing methods for various values of $h$. RNN-OSPG dominates competing methods in this non-Markovian setting across all values of $h$. Also, note the strong outperformance of DNN-OSPG that fits an optimal policy directly vs other Markovian methods (DOS and DNN-FQI) that propagate approximation errors due to recursive estimation of value functions. DOS-ES seeks to adapt to non-Markovian settings by extending its state space with process history. This extension, in turn, explodes the parameter space and training times (see Appendix D, Table 6) while reducing sample efficiency. RRLSM solves this issue by using an RNN but, in turn, trades model flexibility (since the RNN layer is not-trainable). The performance of DOS-ES and RRLSM is quite close in this scenario. Note that even though an RNN is used, the performance of RNN-FQI is significantly worse than RNN-OSPG for positive values of $h$.

**Early-stopping a sequential multi-class classifier:** This is a non-Markovian setting where we are interested in early stopping a sequential multi-class time-series classifier. The longer one delays

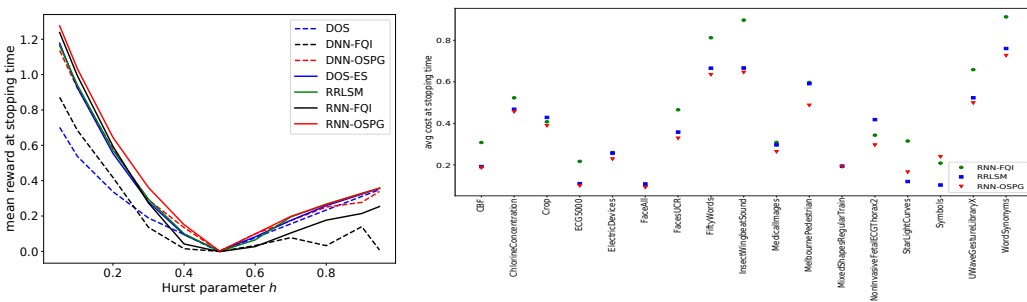

Figure 2: Fractional Brownian motion  Figure 3: Stopping a multi-class classifier

classification, the easier the task becomes since every new time-step reveals information about the class of the series. When deciding on the class of a given series, there is a tradeoff between the number of observations made and classification accuracy. Early stopping of a classifier is an optimal stopping problem with costs (instead of rewards): $\inf_\tau \mathbb{E}[C_\tau]$ using a cost process $\{C_j\}$ with $C_j = [1 - \max_k \mathbb{P}(Y = k|\mathbf{S}_j)] + \alpha \frac{j}{H}$. Here, $Y$ represents the actual class of the time series, with $\mathbf{S}_j$ representing the process history of the time series until time $j$. The first term represents the probability of a classification error at time $j$ while the second term imposes a cost of delaying the classification decision, controlled by $\alpha \in \mathbb{R}^+$. In practice, the true probabilities $\mathbb{P}(Y = k|\mathbf{S}_j)$ are unknown and must be approximated by training a classifier to get a cost model. This approach is related but not equivalent to early classification literature [26, 13, 9, 26, 1] where the goal is to learn both classifier and stopping decision jointly. Since our intent in this paper is to solve the optimal stopping problem, not optimize the cost/reward models, we train a simple baseline RNN classifier that learns parameterized approximations $\mathbb{P}(Y = k|\mathbf{S}_j, \zeta) \approx \mathbb{P}(Y = k|\mathbf{S}_j)\forall j, k$, with parameters $\zeta$. A limitation of this approach is that we are constrained by the quality of the trained classifier. Our classifier is an RNN with a single 20-unit hidden layer connected to a final time-distributed dense layer with softmax activation which produces a probability distribution over classes at each time-step. We train this model to minimize negative log-likelihood (NLL) to produce a calibrated classifier.

There are two possibilities for the cost model. Either we accept the probabilities estimated by the classifier as the true probabilities yielding: $C_j^{est} = [1 - \max_k \mathbb{P}(Y = k|\mathbf{S}_j, \zeta)] + \alpha \frac{j}{H}$, or use empirical miss-classification error, yielding: $C_j^{emp} = \mathbb{I}(Y \neq \arg\max_k \mathbb{P}(Y = k|\mathbf{S}_j, \zeta)) + \alpha \frac{j}{H}$. The latter is a more robust choice and is the evaluation metric used since the stopping decisions will not be blind to classifier calibration errors. However, this setting exposes a significant limitation of value-function methods (backward induction and FQI) since they require the cost model to be available at inference time. Thus, they can only use $C_j^{est}$ and not $C_j^{emp}$, since during inference, empirical errors of the classifier are not revealed until a decision to stop has been made.

We select 17 multi-class time-series classification datasets from the UCR time-series repository [11] (see Appendix D for details of the datasets and selection process). We use 10-random 70%/30% (train/test) data splits to train all models, holding out a 30% of training data as a validation set. Figure 3(b) reports the average stopping costs of the policies learned by the various methods on the test set. RNN-OSPG outperforms backward-induction and FQI methods on 15 of the 17 datasets.

## 6  Conclusions

We introduce a policy gradient algorithm well suited to learning RNN-based optimal stopping policies effective in non-Markovian settings. Our optimal stopping policy gradient (OSPG) algorithm leverages a new Bayesian net formulation of non-Markovian state-action-reward trajectories specific to optimal stopping. This formulation leads to an offline policy gradient algorithm where Bayes net inference eliminates inefficient Monte-Carlo policy rollouts. OSPG outperforms competing methods in non-Markovian settings where existing methods must either deal with the curse of dimensionality (from extended state spaces) or the curse of non-Markovianity (from recursive Markovian approximation). A limitation of OSPG is that since it uses a temporal loss defined over complete trajectories, it may not be suitable for scenarios with only a few extremely long trajectories or when trajectories are partially observed.

## Acknowledgments

We acknowledge Jerry Liu, M. Anthony Lewis, Matt Ellis, Scott Hallworth, and Jennifer Hill at HP Inc. for supporting and sponsoring this work. Finally, we thank all the reviewers for their insightful comments that have helped improve the paper.

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

## Deep Recurrent Optimal Stopping: Supplementary Material

We provide an outline of the supplement that complements our paper.

**Appendix A**: provides proofs of all Theorems in the paper.

**Appendix B**: considers formulations of stopping problems specified in terms of costs instead of rewards and shows how to transform these into a consistent reward formulation.

**Appendix C**: provides discussion and pseudo-code to implement the temporal loss used to learn fitted Q-iteration policies for DNN-FQI and RNN-FQI

**Appendix D**: includes further details and discussion of the experiments, including model hyper-parameter settings, numerical results with confidence intervals, details of compute environment, model sizes, train and inference times, etc. In Appendix E.3, we also include a new experiment that compares OSPG to state-of-the-art PDE benchmarks in the pricing of American options, which is a continuous-time optimal stopping problem.

**Appendix E**: includes a treatment of baseline subtraction in the context of optimal stopping policy gradients (OSPG).

## A    Appendix A: Proofs

### A.1    Proof of Lemma 3.1

*Proof.* The Lemma follows from the Bayes net trajectory model of Figure 1a, and the special structure of OS action trajectories: $A_\tau = 1$ and $A_n = 0, \forall n < \tau$. For any OS state-action trajectory, we have:

$$\mathbb{P}(\mathbf{A}_\tau, \mathbf{S}_H) = \underbrace{\mathbb{P}(\mathbf{S}_0) \prod_{j=1}^{H} \mathbb{P}(\mathbf{S}_j|\mathbf{S}_{j-1})}_{\mathbb{P}(\mathbf{S}_H)} \prod_{n=0}^{\tau} \mathbb{P}(A_n|\mathbf{S}_n) = \mathbb{P}(\mathbf{S}_H) \prod_{n=0}^{\tau} \mathbb{P}(A_n|\mathbf{S}_n) \tag{15}$$

Therefore, conditioning on the state trajectory, we have $\mathbb{P}(\mathbf{A}_\tau|\mathbf{S}_H) = \prod_{n=0}^{\tau} \mathbb{P}(A_n|\mathbf{S}_n)$. Note that by the structure of finite-horizon OS trajectories $\mathbf{A}_\tau$ is a sequence of continue actions terminated by a stop action at $\tau$. Thus there is a bijective mapping between stopping times and complete action trajectories given by:

$$\kappa : \{0, 1, 2, \cdots, H\} \mapsto \{1, 01, 001, \cdots \underbrace{00 \cdots 0}_{H-1 \ 0s} 1\} \tag{16}$$

So $\mathbb{P}(\mathbf{A}_\tau|\mathbf{S}_H) = \mathbb{P}(\kappa(\tau)|\mathbf{S}_H) = \mathbb{P}(\tau|\mathbf{S}_H)$. Consider a trajectory stopping at $\tau = j$ and recall that our stochastic stopping policy is defined as $\phi_j(\mathbf{S}_j) : \mathbb{P}(A_j = 1|\mathbf{S}_j)$.

Thus, if $j = 0$:

$$\mathbb{P}(\tau = 0|\mathbf{S}_H) = \mathbb{P}(\mathbf{A}_0|\mathbf{S}_H) = \mathbb{P}(A_0 = 1|\mathbf{S}_0) = \phi_0(\mathbf{S}_0) \tag{17}$$

If $0 < j < H$:

$$\begin{aligned}
\mathbb{P}(\tau = j|\mathbf{S}_H) &= \mathbb{P}(A_0 = 0, \cdots, A_{j-1} = 0, A_j = 1|\mathbf{S}_H) \\
&= \mathbb{P}(A_j = 1|\mathbf{S}_H) \prod_{n=0}^{j-1} \mathbb{P}(A_n = 0|\mathbf{S}_H) \\
&= \phi_j(\mathbf{S}_j) \prod_{n=0}^{j-1} (1 - \phi_n(\mathbf{S}_n))
\end{aligned} \tag{18}$$

Finally, if $j = H$:

$$
\begin{aligned}
\mathbb{P}(\tau = H | \mathbf{S}_H) &= \mathbb{P}(A_0 = 0, \cdots, A_{H-1} = 0, A_H = 1 | \mathbf{S}_H) \\
&= \mathbb{P}(A_H = 1 | \mathbf{S}_H) \prod_{n=0}^{H-1} \mathbb{P}(A_n = 0 | \mathbf{S}_H) \\
&= \phi_H(\mathbf{S}_H) \prod_{n=0}^{H-1} (1 - \phi_n(\mathbf{S}_n)) \\
&= \prod_{n=0}^{H-1} (1 - \phi_n(\mathbf{S}_n)) \quad (19)
\end{aligned}
$$

where we have used the fact that in the finite horizon setting $\phi_H(\mathbf{S}_H) := 1$ by definition.

We may verify $\sum_{j=0}^{H} \mathbb{P}(\tau = j | \mathbf{S}_H) = 1$ since:

$$
\begin{aligned}
1 - \sum_{j=0}^{H-1} \mathbb{P}(\tau = j | \mathbf{S}_H) &= 1 - \phi_0(\mathbf{S}_0) - \sum_{j=0}^{H-1} \phi_j(\mathbf{S}_j) \prod_{n=0}^{j-1} (1 - \phi_n(\mathbf{S}_n)) \\
&= (1 - \phi_0(\mathbf{S}_0)) \left[ 1 - \phi_1(\mathbf{S}_1) - \sum_{j=1}^{H-1} \phi_j(\mathbf{S}_j) \prod_{n=1}^{j-1} (1 - \phi_n(\mathbf{S}_n)) \right] \\
&= (1 - \phi_0(\mathbf{S}_0))(1 - \phi_1(\mathbf{S}_1)) \times \\
&\qquad \left[ 1 - \phi_2(\mathbf{S}_2) - \sum_{j=2}^{H-1} \phi_j(\mathbf{S}_j) \prod_{n=2}^{j-1} (1 - \phi_n(\mathbf{S}_n)) \right] \\
&\;\;\vdots \\
&= \prod_{n=0}^{H-1} (1 - \phi_n(\mathbf{S}_n)) = \mathbb{P}(\tau = H | \mathbf{S}_H)
\end{aligned}
$$

Further, since $\tau$ is a stopping time random variable we have:

$$
\mathbb{P}(\tau = j | \mathbf{S}_H) = \mathbb{P}(\tau = j | \mathbf{S}_j) := \psi_j(\mathbf{S}_j) \quad (20)
$$

This is because stopping time random variables have the property that $\mathbb{1}(\tau = j)$ is a function of $\mathbf{S}_j$. So we can determine if $\tau = j$ or not by only considering $\mathbf{S}_j$ making the event $\{\tau = j\}$ conditionally independent of $S_{j+k}, \forall k > 0$ given $\mathbf{S}_j$[35]. This completes the proof. $\qquad \square$

## A.2 Proof of Theorem 3.1

*Proof.* We apply Jensen's inequality to the objective $J_{\text{WML}}(\boldsymbol{\theta})$ to obtain a lower bound:

$$
J_{\text{WML}}(\boldsymbol{\theta}) = \sum_{i=1}^{N} \tilde{w}_i \log \sum_{j=0}^{H} \psi_j^{\boldsymbol{\theta}}(\mathbf{s}_{ij}) \tilde{r}_{ij} \geq \sum_{i=1}^{N} \sum_{j=0}^{H} \tilde{w}_i q_{ij} \log \left[ \frac{\psi_j^{\boldsymbol{\theta}}(\mathbf{s}_{ij}) \tilde{r}_{ij}}{q_{ij}} \right] := J(\mathbf{Q}, \boldsymbol{\theta}) \quad (21)
$$

where $\mathbf{Q} = [q_{ij}]$ is any row-stochastic matrix satisfying $q_{ij} \geq 0, \forall i, j$ and $\sum_{j=0}^{H} q_{ij} = 1$. Starting with a given $\boldsymbol{\theta}^{(0)}$ the lower bound is maximized (Jensen's inequality becomes an equality) when $\mathbf{Q} = \mathbf{Q}^{(0)} := [q_{ij}^{(0)}]$ such that:

$$
q_{ij}^{(0)} = \frac{\psi_j^{\boldsymbol{\theta}^{(0)}}(\mathbf{s}_{ij}) \tilde{r}_{ij}}{\sum_{j=0}^{H} \psi_j^{\boldsymbol{\theta}^{(0)}}(\mathbf{s}_{ij}) \tilde{r}_{ij}} \quad (22)
$$

This is the E-step in Theorem 3.1. Note that the E-step does not change the objective, so $J_{\text{WML}}(\boldsymbol{\theta}^{(0)}) = J(\mathbf{Q}^{(0)}, \boldsymbol{\theta})$. By ignoring terms that do not depend on $\boldsymbol{\theta}$, maximizing $J(\mathbf{Q}^{(0)}, \boldsymbol{\theta})$ w.r.t. $\boldsymbol{\theta}$ can be seen as equivalent to maximizing

$$
J_M(\boldsymbol{\theta}) = \sum_{i=1}^{N} \sum_{j=0}^{H} \tilde{w}_i q_{ij} \log \psi_j^{\boldsymbol{\theta}}(\mathbf{s}_{ij}) \quad (23)
$$

This is the M-step. By the lower bound established in equation (21) and since the M-step maximizes $J(\mathbf{Q}^{(0)}, \boldsymbol{\theta})$ to obtain $\boldsymbol{\theta}^{(1)}$, we have $J_{\text{WML}}(\boldsymbol{\theta}^{(1)}) \geq J(\mathbf{Q}^{(0)}, \boldsymbol{\theta}^{(1)}) \geq J(\mathbf{Q}^{(0)}, \boldsymbol{\theta}^{(0)}) = J_{\text{WML}}(\boldsymbol{\theta}^{(0)})$. Therefore a round of E-M results in either an increase or no change in the objective. Since the WML objective is upper-bounded by $N \log H$, the monotone increasing sequence $J_{\text{WML}}(\boldsymbol{\theta}^{(k)})$ converges to a local maximum of the objective. $\qquad\square$

## A.3 Proof of Corollary 3.1.1

*Proof.* It suffices to show that the W-step also increases the objective $J_{\text{WML}}(\tilde{\mathbf{w}}, \boldsymbol{\theta})$.

We maximize $J_{\text{WML}}(\tilde{\mathbf{w}}, \boldsymbol{\theta})$ w.r.t. $\tilde{\mathbf{w}}$, subject to the constraint $\sum_{i=1}^{N} \tilde{w}_i = 1$, resulting the following Lagrangian.

$$\mathcal{L}(\tilde{\mathbf{w}}, \lambda) = \sum_{i=1}^{N} \tilde{w}_i \log \sum_{j=0}^{H} \psi_j^{\boldsymbol{\theta}}(\mathbf{s}_{ij}) \tilde{r}_{ij} - \sum_{i=1}^{N} \tilde{w}_i \log \frac{\tilde{w}_i}{\tilde{r}_i} + \lambda \left(1 - \sum_{i=1}^{N} \tilde{w}_i \right) \tag{24}$$

Taking partial derivative w.r.t. $\tilde{w}_i$ and $\lambda$ and setting to zero, we have:

$$\frac{\partial \mathcal{L}(\tilde{\mathbf{w}}, \lambda)}{\partial \tilde{w}_i} = \log \sum_{j=0}^{H} \psi_j^{\boldsymbol{\theta}}(\mathbf{s}_{ij}) \tilde{r}_{ij} + \log \tilde{r}_i - (1 + \log \tilde{w}_i) - \lambda = 0 \tag{25}$$

$$\frac{\partial \mathcal{L}(\tilde{\mathbf{w}}, \lambda)}{\partial \lambda} = \left(1 - \sum_{i=1}^{N} \tilde{w}_i \right) = 0 \tag{26}$$

Noting that $\tilde{r}_i := \frac{\sum_{j=0}^{H} r_{ij}}{\sum_{i=1}^{N} \sum_{j=0}^{H} r_{ij}}$ and $\tilde{r}_{ij} := \frac{r_{ij}}{\sum_{j=0}^{H} r_{ij}}$ and simplifying, we have:

$$1 + \lambda = \log \left[ \frac{\sum_{j=0}^{H} \psi_j^{\boldsymbol{\theta}}(\mathbf{s}_{ij}) r_{ij}}{r \tilde{w}_i} \right] \tag{27}$$

where $r = \sum_{i=1}^{N} \sum_{j=0}^{H} r_{ij}$. Exponentiation of both sides and cross-multiplication results in:

$$[r \exp(1 + \lambda)] \tilde{w}_i = \sum_{j=0}^{H} \psi_j^{\boldsymbol{\theta}}(\mathbf{s}_{ij}) r_{ij} \tag{28}$$

Summing over $i$, we have:

$$r \exp(1 + \lambda) = \sum_{i=1}^{N} \sum_{j=0}^{H} \psi_j^{\boldsymbol{\theta}}(\mathbf{s}_{ij}) r_{ij} \tag{29}$$

Substituting back in equation (28), we finally have:

$$\tilde{w}_i^* = \frac{\sum_{j=0}^{H} \psi_j^{\boldsymbol{\theta}}(\mathbf{s}_{ij}) r_{ij}}{\sum_{i=1}^{N} \sum_{j=0}^{H} \psi_j^{\boldsymbol{\theta}}(\mathbf{s}_{ij}) r_{ij}} \tag{30}$$

Thus the maximizing $\tilde{\mathbf{w}}^*$ is exactly the W-step. Therefore, the form of the W-step ensures $J_{\text{WML}}(\tilde{\mathbf{w}}^{(k)}, \boldsymbol{\theta}^{(k)}) \geq J_{\text{WML}}(\tilde{\mathbf{w}}^{(k-1)}, \boldsymbol{\theta}^{(k)}) \geq J_{\text{WML}}(\tilde{\mathbf{w}}^{(k-1)}, \boldsymbol{\theta}^{(k-1)})$. Thus we have a monotone increasing and bounded sequence $J_{\text{WML}}(\tilde{\mathbf{w}}^{(k)}, \boldsymbol{\theta}^{(k)})$ converging to a local maximum of $J_{\text{WML}}(\tilde{\mathbf{w}}, \boldsymbol{\theta})$. $\qquad\square$

## A.4 Proof of Theorem 4.1

*Proof.* We use the log-derivative trick [42]. $\nabla_{\boldsymbol{\theta}} \psi_j^{\boldsymbol{\theta}}(\mathbf{s}_j) = \psi_j^{\boldsymbol{\theta}}(\mathbf{s}_j) \nabla_{\boldsymbol{\theta}} \log \psi_j^{\boldsymbol{\theta}}(\mathbf{s}_j)$. So :

$$\nabla_{\boldsymbol{\theta}} J_{OS}(\boldsymbol{\theta}) = \mathbb{E}_{\mathbf{s}_H \sim \mathbb{P}(\mathbf{s}_H)} \left[ \sum_{j=0}^{H} r_j \nabla_{\boldsymbol{\theta}} \psi_j^{\boldsymbol{\theta}}(\mathbf{s}_j) \right] = \mathbb{E}_{\mathbf{s}_H \sim \mathbb{P}(\mathbf{s}_H)} \left[ \sum_{j=0}^{H} r_j \psi_j^{\boldsymbol{\theta}}(\mathbf{s}_j) \nabla_{\boldsymbol{\theta}} \log \psi_j^{\boldsymbol{\theta}}(\mathbf{s}_j) \right] \tag{31}$$

$\qquad\square$

## A.5 Proof of Proposition 4.1

*Proof.* First, for a given $\boldsymbol{\theta}^{(k)}$, substituting for $q_{ij}^{(k)}$ from Theorem 3.1 and $\tilde{w}_i^{(k)}$ from the W-step yields:

$$
\tilde{w}_i^{(k)} q_{ij}^{(k)} = \left[ \frac{\sum_{n=0}^{H} \psi_n^{\boldsymbol{\theta}^{(k)}}(\mathbf{s}_{in}) r_{in}}{\sum_{m=1}^{N} \sum_{n=0}^{H} \psi_n^{\boldsymbol{\theta}^{(k)}}(\mathbf{s}_{mn}) r_{mn}} \right] \left[ \frac{\psi_j^{\boldsymbol{\theta}^{(k)}}(\mathbf{s}_{ij}) \tilde{r}_{ij}}{\sum_{n=0}^{H} \psi_n^{\boldsymbol{\theta}^{(k)}}(\mathbf{s}_{in}) \tilde{r}_{in}} \right] \tag{32}
$$

$$
= \left( \sum_{n=0}^{H} r_{in} \right) \psi_j^{\boldsymbol{\theta}^{(k)}}(\mathbf{s}_{ij}) \tilde{r}_{ij} \left[ \frac{1}{\sum_{m=1}^{N} \sum_{n=0}^{H} \psi_n^{\boldsymbol{\theta}^{(k)}}(\mathbf{s}_{mn}) r_{mn}} \right] \tag{33}
$$

$$
= \underbrace{\left[ \psi_j^{\boldsymbol{\theta}^{(k)}}(\mathbf{s}_{ij}) r_{ij} \right]}_{v_{ij}^{(k)}} \underbrace{\left[ \frac{1}{\sum_{m=1}^{N} \sum_{n=0}^{H} \psi_n^{\boldsymbol{\theta}^{(k)}}(\mathbf{s}_{mn}) r_{mn}} \right]}_{z^{(k)}} \tag{34}
$$

Therefore, we may write the M-step objective as :

$$
J_M^{(k)}(\boldsymbol{\theta}) = z^{(k)} \sum_{i=1}^{N} \sum_{j=0}^{H} v_{ij}^{(k)} \log \psi_j^{\boldsymbol{\theta}}(\mathbf{s}_{ij}) \tag{35}
$$

$$
\propto \frac{1}{N} \sum_{i=1}^{N} \sum_{j=0}^{H} v_{ij}^{(k)} \log \psi_j^{\boldsymbol{\theta}}(\mathbf{s}_{ij}) = \bar{J}_M^{(k)}(\boldsymbol{\theta}) \tag{36}
$$

Dropping the iteration index $(k)$, by defining constants $v_{ij} := \psi_j^{\boldsymbol{\theta}}(\mathbf{s}_{ij}) r_{ij}$ calculated with the most recent value of $\boldsymbol{\theta}$, the M-step objective is equivalent to the following objective:

$$
\bar{J}_M(\boldsymbol{\theta}) = \frac{1}{N} \sum_{i=1}^{N} \sum_{j=0}^{H} v_{ij} \log \psi_j^{\boldsymbol{\theta}}(\mathbf{s}_{ij}) \tag{37}
$$

where $v_{ij} := \psi_j^{\boldsymbol{\theta}}(\mathbf{s}_{ij}) r_{ij}$ is calculated with the most recent value of $\boldsymbol{\theta}$ and held constant. Taking the gradient w.r.t $\boldsymbol{\theta}$ we have

$$
\nabla_{\boldsymbol{\theta}} \bar{J}_M(\boldsymbol{\theta}) = \frac{1}{N} \sum_{i=1}^{N} \sum_{j=0}^{H} v_{ij} \nabla_{\boldsymbol{\theta}} \log \psi_j^{\boldsymbol{\theta}}(\mathbf{s}_{ij}) \tag{38}
$$

Now, substituting for $v_{ij}$, we have:

$$
\nabla_{\boldsymbol{\theta}} \bar{J}_M(\boldsymbol{\theta}) = \frac{1}{N} \sum_{i=1}^{N} \sum_{j=0}^{H} r_{ij} \psi_j^{\boldsymbol{\theta}}(\mathbf{s}_{ij}) \nabla_{\boldsymbol{\theta}} \log \psi_j^{\boldsymbol{\theta}}(\mathbf{s}_{ij}) \tag{39}
$$

This can be expressed as an expectation over sample trajectories as:

$$
\nabla_{\boldsymbol{\theta}} \bar{J}_M(\boldsymbol{\theta}) = \mathbb{E}_{\mathbf{s}_H \sim \mathbb{P}(\mathbf{s}_H)} \left[ \sum_{j=0}^{H} r_j \psi_j^{\boldsymbol{\theta}}(\mathbf{s}_j) \nabla_{\boldsymbol{\theta}} \log \psi_j^{\boldsymbol{\theta}}(\mathbf{s}_j) \right] = \nabla_{\boldsymbol{\theta}} J_{OS}(\boldsymbol{\theta})
$$

Thus the gradient of the transformed M-step objective is identical to the optimal stopping policy gradient (OSPG). Therefore, if we perform the E-step, W-step and a single gradient update, the sequence of policy parameters $\boldsymbol{\theta}_n$ will exactly correspond to the updated OSPG policy parameters. We may therefore appeal to literature on incremental partial M-step E-M algorithms [27] and gradient descent [21] to conclude that for small enough step-size, that increases $\bar{J}_M(\boldsymbol{\theta})$, the policy updates converge to a local maximum of both $J_{WML}(\tilde{\mathbf{w}}, \boldsymbol{\theta})$ and $J_{OS}(\boldsymbol{\theta})$.

□

### A.6 Proof of Corollary 4.1.1

*Proof.* From the proof of Proposition 4.1, the M-step objective is equivalent to the following objective:

$$\bar{J}_M(\boldsymbol{\theta}) = \frac{1}{N} \sum_{i=1}^{N} \sum_{j=0}^{H} v_{ij} \log \psi_j^{\boldsymbol{\theta}}(\mathbf{s}_{ij}) \tag{40}$$

Now, substituting for $\psi_j^{\boldsymbol{\theta}}(\mathbf{s}_{ij})$ in terms of the stopping policy using the trajectory reparameterization lemma (Lemma 3.1):

$$
\begin{aligned}
J_M(\boldsymbol{\theta}) \propto \bar{J}_M(\boldsymbol{\theta}) \ = \ & \frac{1}{N} \sum_{i=1}^{N} v_{i0} \log \phi_0^{\boldsymbol{\theta}}(\mathbf{s}_{i0}) + \sum_{j=1}^{H-1} v_{ij} \left[ \log \phi_j^{\boldsymbol{\theta}}(\mathbf{s}_{ij}) + \sum_{n=0}^{j-1} \log(1 - \phi_n^{\boldsymbol{\theta}}(\mathbf{s}_{in})) \right] \\
& + v_{iH} \sum_{n=0}^{H-1} \log(1 - \phi_n^{\boldsymbol{\theta}}(\mathbf{s}_{in})) \\
= \ & \frac{1}{N} \sum_{i=1}^{N} \left[ \sum_{j=0}^{H-1} v_{ij} \log \phi_j^{\boldsymbol{\theta}}(\mathbf{s}_{ij}) \right] \\
& + v_{i1} \log(1 - \phi_0^{\boldsymbol{\theta}}(\mathbf{s}_{i0})) \\
& + v_{i2} \left[ \log(1 - \phi_0^{\boldsymbol{\theta}}(\mathbf{s}_{i0})) + \log(1 - \phi_1^{\boldsymbol{\theta}}(\mathbf{s}_{i1})) \right] \\
& + v_{i3} \left[ \log(1 - \phi_0^{\boldsymbol{\theta}}(\mathbf{s}_{i0})) + \log(1 - \phi_1^{\boldsymbol{\theta}}(\mathbf{s}_{i1})) + \log(1 - \phi_2^{\boldsymbol{\theta}}(\mathbf{s}_{i2})) \right] \cdots \\
& + v_{iH} \sum_{n=0}^{H-1} \log(1 - \phi_n^{\boldsymbol{\theta}}(\mathbf{s}_{in})) \\
= \ & \frac{1}{N} \sum_{i=1}^{N} \left[ \left[ \sum_{j=0}^{H-1} v_{ij} \log \phi_j^{\boldsymbol{\theta}}(\mathbf{s}_{ij}) \right] + \left[ \sum_{j=0}^{H-1} \log(1 - \phi_j^{\boldsymbol{\theta}}(\mathbf{s}_{ij})) \left\{ \sum_{n=j+1}^{H} v_{in} \right\} \right] \right] \\
= \ & \frac{1}{N} \sum_{i=1}^{N} \sum_{j=0}^{H-1} v_{ij} \log \phi_j^{\boldsymbol{\theta}}(\mathbf{s}_{ij}) + \log(1 - \phi_j^{\boldsymbol{\theta}}(\mathbf{s}_{ij})) \left[ \sum_{n=j+1}^{H} v_{in} \right]
\end{aligned}
$$

Since by Proposition 4.1, we have $\nabla_{\boldsymbol{\theta}} J_{OS}(\boldsymbol{\theta}) = \nabla_{\boldsymbol{\theta}} \bar{J}_M(\boldsymbol{\theta})$. Setting $k_{ij} := \left[ \sum_{n=j+1}^{H} v_{in} \right]$ in the above expression for $\bar{J}_M(\boldsymbol{\theta})$ and taking the gradient, we have:

$$\nabla_{\boldsymbol{\theta}} J_{OS}(\boldsymbol{\theta}) = \frac{1}{N} \sum_{i=1}^{N} \sum_{j=0}^{H} \left[ \frac{v_{ij}(1 - \phi_j^{\boldsymbol{\theta}}(\mathbf{s}_{ij})) - k_{ij} \phi_j^{\boldsymbol{\theta}}(\mathbf{s}_{ij})}{\phi_j^{\boldsymbol{\theta}}(\mathbf{s}_{ij})(1 - \phi_j^{\boldsymbol{\theta}}(\mathbf{s}_{ij}))} \right] \nabla_{\boldsymbol{\theta}} \phi_j^{\boldsymbol{\theta}}(\mathbf{s}_{ij}) \tag{41}$$

This completes the proof of Corollary 4.1.1

$\square$

## B Dealing with costs instead of rewards

Although one could have used the negative of cost as a reward, this is inconsistent with our Bayesian net model since we require positive rewards to define the reward augmented trajectory model. The following result addresses this issue by transforming a problem with costs to one with a suitable reward specification.

**Proposition B.1.** Given costs $c_{ij} \geq 0$ the following two problems are equivalent:

$$\arg\min_{\boldsymbol{\theta}} \frac{1}{N} \sum_{i=1}^{N} \sum_{j=0}^{H} c_{ij} \psi_j^{\boldsymbol{\theta}}(\mathbf{s}_{ij}) \equiv \arg\max_{\boldsymbol{\theta}} \frac{1}{N} \sum_{i=1}^{N} \sum_{j=0}^{H} \underbrace{\left( \sum_{n=0}^{H} c_{in} \right) \left( 1 - \frac{c_{ij}}{\sum_{n=0}^{H} c_{in}} \right)}_{r'_{ij}} \psi_j^{\boldsymbol{\theta}}(\mathbf{s}_{ij})$$

*Proof.* Starting with the cost minimization problem, we may write:

$$\arg\min_{\boldsymbol{\theta}} \frac{1}{N}\sum_{i=1}^{N}\sum_{j=0}^{H} c_{ij}\psi_j^{\boldsymbol{\theta}}(\mathbf{s}_{ij}) = \arg\min_{\boldsymbol{\theta}} \frac{1}{N}\sum_{i=1}^{N} c_i \sum_{j=0}^{H} \tilde{c}_{ij}\psi_j^{\boldsymbol{\theta}}(\mathbf{s}_{ij}) \tag{42}$$

where $c_i := \sum_{j=0}^{H} c_{ij}$ and $\tilde{c}_{ij} = \frac{c_{ij}}{c_i}$. Since $\sum_{j=0}^{H} \psi_j^{\boldsymbol{\theta}}(\mathbf{s}_{ij}) = 1$, we may subtract the constant $\frac{1}{N}\sum_{i=1}^{N} c_i \sum_{j=0}^{H} \psi_j^{\boldsymbol{\theta}}(\mathbf{s}_{ij})$ from the objective yielding:

$$
\begin{aligned}
\arg\min_{\boldsymbol{\theta}} \frac{1}{N}\sum_{i=1}^{N}\sum_{j=0}^{H} c_{ij}\psi_j^{\boldsymbol{\theta}}(\mathbf{s}_{ij}) &= \arg\min_{\boldsymbol{\theta}} \frac{1}{N}\sum_{i=1}^{N} c_i \sum_{j=0}^{H} \left(\tilde{c}_{ij} - 1\right)\psi_j^{\boldsymbol{\theta}}(\mathbf{s}_{ij}) \\
&= \arg\max_{\boldsymbol{\theta}} \frac{1}{N}\sum_{i=1}^{N} c_i \sum_{j=0}^{H} \left(1 - \tilde{c}_{ij}\right)\psi_j^{\boldsymbol{\theta}}(\mathbf{s}_{ij}) \\
&= \arg\max_{\boldsymbol{\theta}} \frac{1}{N}\sum_{i=1}^{N}\sum_{j=0}^{H} \underbrace{\left(\sum_{n=0}^{H} c_{in}\right)\left(1 - \frac{c_{ij}}{\sum_{n=0}^{H} c_{in}}\right)}_{r_{ij}'}\psi_j^{\boldsymbol{\theta}}(\mathbf{s}_{ij})
\end{aligned}
$$

where we have expanded $c_i$ and $\tilde{c}_{ij}$. This completes the proof. $\qquad\square$

## C Neural Fitted Q-iteration approaches: DNN-FQI and RNN-FQI

Fitted Q-iteration methods [39, 40] use the Wald-Bellman equation (WBE) as follows: First, given parameterized function approximations $K_j^{\boldsymbol{\theta}}(S_j)$, a single Wald-Bellman step is bootstrapped, for a batch of trajectories yielding $\hat{V}_j(s_{ij}) = \max\{r_{ij}, K_j^{\boldsymbol{\theta}}(s_{ij})\}, \forall j < H, 1 \le i \le N$. Next the parameters of the continuation function are fit: $\boldsymbol{\theta}^* = \arg\min \sum_i \sum_j (K_j^{\boldsymbol{\theta}}(s_{ij}) - \hat{V}_{j+1}(s_{ij+1}))^2$, and the process is iterated. To provide competent DNN/RNN baselines for fitted Q-iteration (FQI) methods that are missing in the literature, we introduce a temporal FQI loss (Algorithm 2).

---

**Algorithm 2** Pseudo-code for mini-batch computation of the FQI loss

---

    **Input:** $\mathbf{R} := [r_{ij}]$, $\mathbf{K} := \left[K_j^{\boldsymbol{\theta}}(\mathbf{s}_{ij})\right]$ {$1 \le i \le N_b, 0 \le j \le H$, $N_b$ is batch size}
    $\mathbf{V}_{:,0:H-1} = \text{stop-gradient}(\max\{\mathbf{R}_{:,0:H-1}, \mathbf{K}_{:,0:H-1}\})$ {bootstrap WBE to get value targets}
    $\mathbf{V}_{:,H} = \mathbf{R}_{:,H}$ {final target is reward for last step}
    $\text{J} = \text{MSE}(\mathbf{V}_{:,1:H}, \mathbf{K}_{0,0:H-1})$ {next step value is target for current continuation function}

---

## D Further details of the experiments

All experiments were performed on a shared server configured with $2 \times$ Intel Xeon Silver 12-core, 2.10 GHz CPUs with 256GB RAM and equipped with 6 NVIDIA 2080Ti GPUs. However, experiments were run on a single GPU at a time and no computation was distributed across GPUs.

### D.1 Model hyper-parameter settings

Table 2 shows general hyper-parameter settings used for all experiments. We apply Batch normalization to the input and outputs of layer activation functions at all hidden layers. Due to the dense correlation structure between assets at each time-step of the American option pricing experiment, we choose the hidden units to be greater than the input dimension $d$.

As with RL policy gradients, we may subtract a baseline value[29] to reduce variance. The OSPG algorithm uses baseline $b$ that does not depend on time-index $j$, sufficient to guarantee an unbiased OSPG estimator (see Appendix E). In our experiments, we use:

$$b = \frac{1}{NH}\sum_{i=1}^{N}\sum_{j=0}^{H} r_{ij} \tag{43}$$

Table 2: model hyper-parameter settings

| method | hyper-parameter | tuned range | value |
|---|---|---|---|
| DOS, DNN-FQI, DNN-OSPG | num hidden layers | n/a | 2 |
| RRLSM, RNN-FQI, RNN-OSPG | num hidden layers | n/a | 1 |
| All models | hidden layer units | n/a | 20 |
| All models (Am. Option pricing) | hidden layer units | n/a | $20 + d$ |
| All models | batch size | n/a | 64 |
| All models (Am. Option pricing) | batch size | n/a | 128 |
| All models | learning rate | $\{0.01, 0.001, 0.0001\}$ | 0.001 |
| All models | epochs | early stopping | 100 |
| All models | batches/epoch | n/a | 200 |
| All models | optimizer | n/a | Adam |
| RRLSM | Kernel noise std | n/a | 0.0001 |
| RRLSM | Recurrent noise std | n/a | 0.3 |

## D.2  Pricing Bermudan options

Table 3 shows model sizes (in trainable parameters), training, and inference times (per time-step). Model sizes grow with input dimension except for RRLSM, which uses an RNN with random, non-trainable weights to extract features from the input. The parameter size of DOS is about an order of magnitude higher than DNN-FQI and DNN-OSPG since parameters in backward induction methods like DOS grow linearly with the number of time-steps (parameters are not shareable across time-steps). Training and inference times are also high since individual models must be fit and inferred at each time-step.

Table 3: model sizes and compute times for Bermudan max-call experiment

| method | assets | model-size (params) | mean training time (seconds) | mean time/prediction ($\mu$-seconds) |
|---|---|---|---|---|
| DOS | 20 | 9,225 | 271 | 3 |
| DNN-FQI | 20 | 1,047 | 23 | 5 |
| DNN-OSPG | 20 | 1,047 | 29 | 7 |
| RRLSM | 20 | 198 | 2 | 14 |
| RNN-FQI | 20 | 2,807 | 32 | 8 |
| RNN-OSPG | 20 | 2,807 | 37 | 8 |
| DOS | 50 | 15,165 | 289 | 3 |
| DNN-FQI | 50 | 1,707 | 28 | 6 |
| DNN-OSPG | 50 | 1,707 | 29 | 7 |
| RRLSM | 50 | 198 | 2 | 14 |
| RNN-FQI | 50 | 4,667 | 32 | 8 |
| RNN-OSPG | 50 | 4,667 | 32 | 8 |
| DOS | 100 | 25,065 | 326 | 3 |
| DNN-FQI | 100 | 2,807 | 30 | 6 |
| DNN-OSPG | 100 | 2,807 | 28 | 6 |
| RRLSM | 100 | 198 | 2 | 14 |
| RNN-FQI | 100 | 7,767 | 33 | 8 |
| RNN-OSPG | 100 | 7,767 | 30 | 8 |
| DOS | 200 | 44,865 | 317 | 3 |
| DNN-FQI | 200 | 5,007 | 30 | 6 |
| DNN-OSPG | 200 | 5,007 | 27 | 6 |
| RRLSM | 200 | 198 | 2 | 15 |
| RNN-FQI | 200 | 13,967 | 35 | 8 |
| RNN-OSPG | 200 | 13,967 | 30 | 8 |

Table 4: American geometric-call option pricing: Results

| | | | average return (error %) | | | | |
|---|---|---|---|---|---|---|---|
| $d$ | $s_0$ | $p^*$ | LS [23] | PDE-DGM [37] | PDE-BSDE [7] | DNN-FQI [34] | DNN-OSPG |
| 7 | 90 | 5.9021 | 5.8440 (0.98%) | NA | **5.8822** (0.34%) | 5.7977 (1.77%) | 5.8704 (0.54%) |
| 7 | 100 | 10.2591 | 10.1736 (0.83%) | NA | 10.2286 (0.30%) | 10.1022 (1.53%) | **10.2518** (0.07%) |
| 7 | 110 | 15.9878 | 15.8991 (0.55%) | NA | **15.9738** (0.09%) | 15.0487 (5.87%) | 15.9699 (0.11%) |
| 13 | 90 | 5.7684 | 5.5962 (3.00%) | NA | **5.7719** (0.06%) | 5.7411 (0.47%) | 5.7436 (0.43%) |
| 13 | 100 | 10.0984 | 9.9336 (1.60%) | NA | **10.1148** (0.16%) | 9.9673 (1.30%) | 10.0691 (0.29%) |
| 13 | 110 | 15.8200 | 15.6070 (1.40%) | NA | **15.8259** (0.04%) | 14.7759 (6.60%) | 15.8107 (0.06%) |
| 20 | 90 | 5.7137 | 5.2023 (9.00%) | NA | **5.7105** (0.06%) | 5.6607 (0.93%) | 5.6983 (0.27%) |
| 20 | 100 | 10.0326 | 9.5964 (4.40%) | **10.0296** (0.03%) | 10.0180 (0.15%) | 9.6372 (3.94%) | 10.0100 (0.23%) |
| 20 | 110 | 15.7513 | 15.2622 (3.10%) | NA | 15.7425 (0.06%) | 14.9345 (5.19%) | **15.7553** (0.03%) |
| 100 | 90 | 5.6322 | OOM | NA | 5.6154 (0.30%) | 5.3858 (4.38%) | **5.6211** (0.20%) |
| 100 | 100 | 9.9345 | OOM | **9.9236** (0.11%) | 9.9187 (0.16%) | 9.3954 (5.43%) | 9.8954 (0.40%) |
| 100 | 110 | 15.6491 | OOM | NA | 15.6219 (0.17%) | 14.6335 (6.49%) | **15.6301** (0.12%) |
| 200 | 100 | 9.9222 | OOM | 9.9004 (0.22%) | **9.9088** (0.14%) | 9.3772 (5.49%) | 9.8991 (0.23%) |

## D.3 Pricing American options

The scope of the paper is solving *discrete-time, finite-horizon, model-free* optimal stopping problems. Bermudan options that have discrete exercise opportunities are one example application. American options, which are more popular, are based on continuous-time asset price evolution and have a continuum of possible exercise times. One way to convert this to our discrete-time setting is to solve related Bermudan options. These options limit exercise opportunities to a fine discrete time grid.

State-of-the-art algorithms for pricing American options are based on Partial differential equation (PDE) methods. These methods are model-based since they start with a PDE (such as the multi-dimensional Black-Scholes Model) defining process evolution. For example, PDE methods often assume Markovian Black-Scholes Dynamics, and the PDEs to be solved require the Black-Scholes model parameters, such as covariance of the Brownian motion, volatility, risk-free interest rate, and dividend yield. In contrast, model-free methods, such as FQI and OSPG algorithms, do not use prior information on the evolution dynamics of the underlying stochastic process.

Nevertheless, we compare our model-free OSPG method against state-of-the-art PDE methods such as the Deep Galerkin Method (DGM) [37] and Backward Stochastic Differential Equations method (BSDE)[7] by suitable discretization of the original continuous time-problem. Note that the PDE methods also require discretization of the original PDE (ex, using the Euler-Maruyama scheme) or random sampling (as used in DGM) but do not end up directly solving a Bermudan option.

We consider multi-dimensional continuous-time American geometric-average call options with Black-Scholes dynamics considered in [37] and [7]. The payoff of these options depends on the price of $d$ underlying assets with multi-dimensional Black-Scholes dynamics and the strike price $K$. The dynamics are Markovian, with payoff (reward) given by:

$$S_t^m = s_0^m \exp([r - \delta_m - \sigma_m^2/2]t + \sigma W_t^m, \quad R_t = \left( \left[ \prod_{m=1}^d S_t^m \right]^{\frac{1}{d}} - K \right)^+ \quad (44)$$

for $m = 1, 2, \cdots d$. $r \in \mathbb{R}$ is the risk-free rate, $s_0^m \in (0, \infty)$ represents the initial price, $\delta_m \in [0, \infty)$ is the dividend yield, $\sigma_m \in (0, \infty)$ the volatility and $W$ is a $d$-dimensional Brownian motion, with instantaneous correlation between its $d$ components given by $\mathbb{E}[W_t^i W_t^j] = \rho_{ij}t$. The reward for exercise at time $t$ is given by $R_t$. We discretize $t$ into $H + 1 = 100$ possible exercise opportunities, using times $t_j = jT/H$ for $j = 0, 1, \cdots H$. $T$ is the option expiration duration in years. This yields the stopping problem: $\sup_{0 \leq \tau \leq H} \mathbb{E}[R_\tau]$.

The specific option we consider is characterized by the following parameters: $K = 100$, $r = 0.0$, $\sigma_m = 0.25$, $\rho_{ij} = 0.75$ $\forall i \neq j$, $\delta_m = 0.02$, $T = 2$. The exact price of this option, $p^*$, can be determined semi-analytically for comparison [7]. We generate 10,000 batches (with a batch size of 128) for training and compute option prices on a 3,000-batch test sample. We compare vs. published results from state-of-the-art model-based PDE baselines, including the Deep Galerkin Method (PDE-DGM) [37], the Backward Stochastic Differential Equation (PDE-BSDE) method [7]. We also include published results [7] from the industry standard Longstaff-Schwartz (LS) option

Table 5: Stopping a fractional Brownian motion: Results

| $h$ | average return (standard deviation) | | | | | | |
|---|---|---|---|---|---|---|---|
| | DOS | DNN-FQI | DNN-OSPG | DOS-ES | RRLSM | RNN-FQI | RNN-OSPG |
| 0.05 | 0.70 (0.01) | 0.87 (0.06) | 1.14 (0.00) | 1.18 (0.01) | 1.16 (0.00) | 1.24 (0.04) | **1.28** (0.01) |
| 0.10 | 0.54 (0.01) | 0.68 (0.04) | 0.92 (0.01) | 0.93 (0.01) | 0.94 (0.00) | 0.99 (0.03) | **1.03** (0.02) |
| 0.20 | 0.34 (0.01) | 0.42 (0.08) | 0.58 (0.00) | 0.55 (0.01) | 0.57 (0.00) | 0.59 (0.04) | **0.64** (0.01) |
| 0.30 | 0.19 (0.00) | 0.14 (0.07) | 0.29 (0.10) | 0.28 (0.01) | 0.29 (0.00) | 0.27 (0.08) | **0.36** (0.01) |
| 0.40 | 0.10 (0.01) | 0.02 (0.03) | 0.13 (0.00) | 0.09 (0.01) | 0.10 (0.00) | 0.04 (0.04) | **0.15** (0.00) |
| 0.50 | **0.00** (0.00) | **0.00** (0.00) | **0.00** (0.00) | **0.00** (0.00) | **0.00** (0.00) | **0.00** (0.00) | **0.00** (0.00) |
| 0.60 | 0.08 (0.01) | 0.03 (0.03) | **0.10** (0.00) | 0.08 (0.01) | 0.06 (0.00) | 0.03 (0.03) | **0.10** (0.03) |
| 0.70 | 0.16 (0.01) | 0.08 (0.08) | 0.17 (0.02) | 0.18 (0.01) | 0.19 (0.00) | 0.10 (0.07) | **0.20** (0.00) |
| 0.80 | 0.23 (0.01) | 0.03 (0.05) | 0.25 (0.00) | 0.26 (0.01) | **0.27** (0.00) | 0.18 (0.09) | **0.27** (0.01) |
| 0.90 | 0.31 (0.01) | 0.14 (0.12) | 0.28 (0.10) | 0.32 (0.01) | **0.33** (0.00) | 0.21 (0.12) | **0.33** (0.00) |
| 0.95 | 0.35 (0.01) | 0.01 (0.05) | 0.34 (0.01) | **0.36** (0.00) | **0.36** (0.00) | 0.25 (0.08) | **0.36** (0.00) |

pricing algorithm [23], which uses linear approximation with multi-dimensional basis functions, suitable for low dimensional settings.

Table 4 summarizes the experiment's results. Published results [7] for LS, DGM, and BSDE are included. NA denotes results not reported in the literature, while OOM indicates *out of memory*. Our model-free DNN-OSPG algorithm compares favorably with state-of-the-art model-based PDE-based option pricing methods that have prior knowledge of process evolution. Specifically, unlike LS, whose accuracy degrades with the number of correlated assets, and DNN-FQI, whose accuracy degrades up to 6.5%, DNN-OSPG retains excellent performance (< 0.55% error) closely approaching the model-based PDE methods (< 0.35% error). Of course, a key advantage of DNN-OSPG is that it can be used to price options for which no known PDE is available to describe process evolution.

## D.4 Stopping a fractional Brownian motion

Table 5 provides numerical values corresponding to Figure 2. RNN-OSPG dominates competing methods across all values the Hurst parameter in this non-Markovian setting.

Table 6: Stopping a fractional Brownian motion: Model sizes and compute times

| method | model-size (params) | mean training time (seconds) | mean time/prediction ($\mu$-seconds) |
|---|---|---|---|
| DOS | 62,900 | 2250 | 26.9 |
| DNN-FQI | 651 | 21 | 0.5 |
| DNN-OSPG | 651 | 29 | 0.5 |
| DOS-ES | 276,300 | 2228 | 25.9 |
| RRLSM | 2,200 | 9 | 5.0 |
| RNN-FQI | 1,691 | 14 | 1.1 |
| RNN-OSPG | 1,691 | 79 | 1.1 |

## D.5 Early stopping a sequential multi-class classifier

We start with the 34 datasets, and corresponding values of $\alpha$ used in the [1] and remove binary classification datasets. This leaves 17 datasets. However, many of these have very few series or have a large number of classes relative to series size, which might make them unsuitable for training RNN classifiers. Nevertheless, we report results on all 17 datasets. Table 7 reports the experiment's mean and standard deviation (over ten random splits) results over 17 UCR time-series datasets. Figure 2 provides a graphical visualization of the same results. $N$ denotes the number of trajectories, $H$ denotes the length of the series, and $K$ denotes the number of classes. $\alpha$ trades-off classification accuracy and earliness. RNN-OSPG achieves the best performance on 15 of the 17 datasets.

Table 7: Early stopping a sequential multi-class classifier: Results

| Dataset | N/H/K/$\alpha$ | average cost (standard deviation) | | |
|---|---|---|---|---|
| | | RRLSM | RNN-FQI | RNN-OSPG |
| CBF | 930/128/3/0.8 | 0.192 (0.017) | 0.308 (0.138) | **0.186** (0.011) |
| ChlorineConcentration | 4,307/166/3/0.4 | 0.468 (0.011) | 0.523 (0.073) | **0.455** (0.003) |
| Crop | 24,000/46/24/0.06 | 0.428 (0.005) | 0.407 (0.010) | **0.389** (0.008) |
| ECG5000 | 5000/140/5/0.5 | 0.110 (0.005) | 0.217 (0.059) | **0.099** (0.006) |
| ElectricDevices | 16,637/96/7/0.1 | 0.257 (0.015) | 0.260 (0.012) | **0.228** (0.009) |
| FaceAll | 2,250/131/14/0.01 | 0.108 (0.025) | 0.100 (0.010) | **0.092** (0.017) |
| FaceUCR | 2,250/131/14/0.5 | 0.358 (0.035) | 0.465 (0.049) | **0.328** (0.028) |
| FiftyWords | 905/270/50/0.5 | 0.667 (0.027) | 0.812 (0.041) | **0.634** (0.029) |
| InsectWingbeatSound | 2,200/256/11/1 | 0.666 (0.064) | 0.897 (0.110) | **0.646** (0.080) |
| MedicalImages | 1,141/99/10/0.07 | 0.296 (0.033) | 0.309 (0.031) | **0.263** (0.038) |
| MelbournePedestrian | 3,633/24/10/0.8 | 0.591 (0.065) | 0.597 (0.068) | **0.487** (0.044) |
| MixedShapesRegularTrain | 2,925/1024/5/0.1 | 0.194 (0.024) | 0.194 (0.008) | **0.191** (0.037) |
| NonInvasiveFetalECGThorax2 | 3,765/750/42/0.04 | 0.418 (0.094) | 0.343 (0.008) | **0.295** (0.064) |
| StarLightCurves | 9,236/1024/3/0.3 | **0.120** (0.043) | 0.315 (0.067) | 0.165 (0.035) |
| Symbols | 1,020/398/6/0.2 | **0.103** (0.013) | 0.208 (0.030) | 0.239 (0.070) |
| UWaveGestureLibraryX | 4,478/315/8/0.5 | 0.523 (0.023) | 0.658 (0.026) | **0.498** (0.040) |
| WordSynonyms | 905/270/25/0.6 | 0.760 (0.024) | 0.913 (0.043) | **0.727** (0.041) |

# E   Baseline subtraction for variance reduction

As with RL policy gradients, we may subtract a baseline value[29] to reduce variance. However, unlike the general RL case, due to the stopping-time-based formulation of the OSPG, the OSPG baseline should not be time-varying.

**Proposition E.1** (baseline subtraction for OSPG). The optimal stopping policy gradient (OSPG) of Theorem 4.1 is invariant to the subtraction of a constant (w.r.t. trajectory) baseline from every reward in the trajectory. Thus:

$$
\begin{aligned}
\nabla_{\boldsymbol{\theta}} J_{OS}(\boldsymbol{\theta}) &= \mathbb{E}_{\mathbf{s}_H \sim \mathbb{P}(\mathbf{s}_H)} \left[ \sum_{j=0}^{H} r_j \psi_j^{\boldsymbol{\theta}}(\mathbf{s}_j) \nabla_{\boldsymbol{\theta}} \log \psi_j^{\boldsymbol{\theta}}(\mathbf{s}_j) \right] \\
&= \mathbb{E}_{\mathbf{s}_H \sim \mathbb{P}(\mathbf{s}_H)} \left[ \sum_{j=0}^{H} (r_j - b) \psi_j^{\boldsymbol{\theta}}(\mathbf{s}_j) \nabla_{\boldsymbol{\theta}} \log \psi_j^{\boldsymbol{\theta}}(\mathbf{s}_j) \right]
\end{aligned}
$$

*Proof.* It suffices to show $\mathbb{E}_{\mathbf{s}_H \sim \mathbb{P}(\mathbf{s}_H)} \left[ \sum_{j=0}^{H} b \psi_j^{\boldsymbol{\theta}}(\mathbf{s}_j) \nabla_{\boldsymbol{\theta}} \log \psi_j^{\boldsymbol{\theta}}(\mathbf{s}_j) \right] = 0$. We proceed as follows:

$$
\begin{aligned}
E_{\mathbf{s}_H \sim \mathbb{P}(\mathbf{s}_H)} \left[ \sum_{j=0}^{H} b \psi_j^{\boldsymbol{\theta}}(\mathbf{s}_j) \nabla_{\boldsymbol{\theta}} \log \psi_j^{\boldsymbol{\theta}}(\mathbf{s}_j) \right] &= E_{\mathbf{s}_H \sim \mathbb{P}(\mathbf{s}_H)} \left[ \sum_{j=0}^{H} b \psi_j^{\boldsymbol{\theta}}(\mathbf{s}_j) \frac{\nabla_{\boldsymbol{\theta}} \psi_j^{\boldsymbol{\theta}}(\mathbf{s}_j)}{\psi_j^{\boldsymbol{\theta}}(\mathbf{s}_j)} \right] \\
&= E_{\mathbf{s}_H \sim \mathbb{P}(\mathbf{s}_H)} \left[ \nabla_{\boldsymbol{\theta}} \sum_{j=0}^{H} b \psi_j^{\boldsymbol{\theta}}(\mathbf{s}_j) \right] \\
&= E_{\mathbf{s}_H \sim \mathbb{P}(\mathbf{s}_H)} \left[ \nabla_{\boldsymbol{\theta}} b \underbrace{\sum_{j=0}^{H} \psi_j^{\boldsymbol{\theta}}(\mathbf{s}_j)}_{=1} \right] \\
&= 0 \quad\quad\quad (45)
\end{aligned}
$$

$\square$

