# OpenReview forum: "Deep Recurrent Optimal Stopping"
_NeurIPS.cc/2023/Conference — NeurIPS 2023 poster_

### Official Review · Reviewer_vhpm · 2023-07-02

**Soundness:** 3 good
**Presentation:** 2 fair
**Contribution:** 2 fair
**Rating:** 6
**Confidence:** 3

**Summary:**

The paper purports to develop a framework of optimal stopping that generalizes the previous approaches by incorporating non-Markovian settings and using a Bayesian network formulation.

**Strengths:**

The only strength of this paper in this reviewer's opinion is the fact that it tackles an important problem.

**Weaknesses:**

The paper is written in a manner that makes it very difficult to discern its contents. The presentation and the proofs are ridiculously long-winded even though they could they be easily written in much more succinct manner. Although the reformulation of the optimal stopping problem as a Bayesian is interesting at first sight, I don't see how it is useful unfortunately, especially given the fact that I have concerns about they are even maximizing. See the next section for more concrete examples of the problems this paper has.

**Questions:**

1. It isn’t clear why every finite stopping time (as defined in the literature as a measurable function $\tau \colon \Omega \to \mathbb{N}$ such that $\\{\tau \le n\\} \in \mathcal{F} _ n$ for all $n \in \mathbb N$; here and elsewhere in the review $(\mathcal{F})_{n \in \mathbb N}$ is the appropriate filtration which makes the relevant stochastic processes adapted) can be written in the form of Definition 2.3. I don’t believe it is even true — for intuitively the same reason why there are stopping times that are not hitting times. This makes me doubt that the paper is solving the desired optimal stopping problem and is limited to a rather special case.
2. What does the notation $\arg \sup$ in Definition 2.4 mean? It isn’t immediately obvious that a optimal stopping time $\tau^*$ will exist and therefore a small note showing this would be helpful. Perhaps a mention of the classical result that a finite optimal stopping time exists if and only if $\tau_0 := \inf \\{n \in \mathbb N : U_n = R_n\\} < \infty$ a.s., where $U_n := \text{ess sup} _ {\tau \in \mathcal{T} _ n} \mathbb{E}[R_\tau \mid \mathcal{F}_n]$ and $\mathcal{T}_n$ being the collection of all finite stopping times $\tau$ satisfying $\tau \ge n$, would make this immediate. (Note: it can be shown that $U_n = V_n$ as defined in the paper.)
3. As mentioned above $V_j$ should be

    $$
    V_j(\mathbf S_j) = \text{ess sup}_{\tau \in \mathcal{T}_j} \mathbb{E}[R_\tau \mid \mathcal{F}_j],
    $$

    which is a *random variable* (see https://almostsuremath.com/2019/01/06/essential-suprema/ for the definition of essential supremum)*.* This isn’t what equation (1) says. Even the Wald-Bellman equation as written in the paper says that $V_j(\mathbf S_j)$ is a random variable.

**Limitations:**

Not applicable.

---

> ### Author Rebuttal · Authors · 2023-08-09
>
> We appreciate the reviewer's concerns regarding the definitions in the paper. Our treatment is closely related to the approach in the following references.
>
> **[27]** A. N. Shiryaev, "Stochastic Disorder Problems"
>
> **[Poor]** H. Vincent Poor, "An Introduction to Signal Detection and Estimation"
>
> **[Fischer]** Fischer, Tom. “Stopping times are hitting times: a natural representation.” Statistics & Probability Letters (2011)
>
> We hope that the reviewer reconsiders the rating of the paper, since there were no issues other than clarifications w.r.t. definitions. Specific responses are as follows.
>
> > The reviewer expressed doubts about the definition of the optimal stopping problem in the paper, specifically, the stated concerns are:
> **a)** *It isn’t clear why every finite stopping time can be written in the form of Definition 2.3.* **b)** *I don’t believe it is even true-for intuitively the same reason  why there are stopping times that are not hitting times. This makes me doubt that the paper is solving the desired optimal stopping problem and is limited to a rather special case.*
>
> **Response:**  We clarify that the  formulation of the optimal stopping time in definition 2.3 applies to general discrete-time, finite horizon setting and is not a special case. Specific responses to a) and b) are as follows
>
> **a)** Definition 2.3 in our manuscript is the same as that in the classic textbook [Poor] "An Introduction to Signal Detection and Estimation" by H. Vincent Poor. Please see both page 137 and page 145 where the **policy stopping time** is defined as :
>
> $N(\phi) = \min$  \{ $n | \phi_n (Y_1, Y_2, \cdots, Y_n) = 1$ \}
>
> Here $N(\phi)$ is the policy stopping time,  $\phi$ is equivanent to our stopping policy $\varphi$, and $Y_1, Y_2, \cdots, Y_n$ is equivalent to ${\bf S}_j$ in our notation.
>
> **b)** As the reviewer remarks, hitting times are stopping times (by the Debut Theorem).
>
> However, on a technical note, stopping times can also be interpreted as hitting times **w.r.t. a "stopping process"**, see for example[Fischer] para 2 on page 1: *"Astonishingly, it seems to be less widely taught (and maybe known) that the inverse is true as well: for any stopping time there exists an adapted stochastic process and a Borel measurable set such that the corresponding hitting time will be exactly this stopping time"*.
>
> > What does the notation in Definition 2.4 mean? It isn’t immediately obvious that a optimal stopping time will exist and therefore a small note showing this would be helpful.
>
> **Response:** The $\arg \sup$ notation for $\tau^*$  in definition 2.4 simply denotes that the optimal stopping time (if one exists) satisfies $\mathbb{E} [ R_{\tau^*} ] = \sup_{\tau} \mathbb{E} [ R_{\tau} ]$.
>
> Existence and finiteness of $\tau^*$ is guaranteed if $\mathbb{E} \sup_{k\geq 0} | R_{k}| < \infty$. See page 58 of [27] A. N. Shiryaev, "Stochastic Disorder Problems" book. We will add a short note regarding this technical condition in the revised paper as suggested by the reviewer. We agree that the condition suggested by the reviewer is indeed a sound alternative condition. However, our treatment is along the lines of [27] and easy to verify.
>
> >Use ess sup in equation (1) to define $V_j$. It is a random variable unlike what equation (1) suggests.
>
> **Response:** The issue is due to a typo in equation (1): It should be
>
>  $\mathbb{E} [ V_j({\bf S_{\textit j}}) ] = \sup_{\tau \geq j} \mathbb{E} [ R_{\tau} ]$
>
>  instead of
>
>  $V_j({\bf S_{\textit j}}) = \sup_{\tau \geq j} \mathbb{E} [ R_{\tau} ]$.
>
> This result appears on page 60, Theorem 1 (case of a finite time horizon)  of [27] A. N. Shiryaev, "Stochastic Disorder Problems" book.

---

> > ### Comment · Reviewer_vhpm · 2023-08-18
> >
> > 1. Thank you very much for the reference to the Fischer's paper. It is a very interesting result! I did not know about this. In light of this I have significantly revised my rating for the paper.
> >
> > 2. The existence of $\tau^*$ is _not_ guaranteed by the condition $\mathbb{E} \sup_{k \ge 0} |R_k| < \infty$. Shiryaev on page 58 only says that this condition is sufficient for the existence and finiteness of $\sup_{\tau \in \mathfrak{M}} \mathbb{E} G_\tau$. It doesn't say anything about the existence of $\tau^*$.

---

> > > ### Author Response · Authors · 2023-08-19
> > >
> > > We thank you for your valuable comments and kind reconsideration of our paper.  In response to your following point:
> > >
> > > > The existence of $\tau^*$ is not guaranteed by the condition $\mathbb{E} \sup_{k \ge 0} |R_k| < \infty$. Shiryaev on page 58 only says that this condition is sufficient for the existence and finiteness of $\sup_{\tau \in \mathfrak{M}} \mathbb{E} G_\tau$. It doesn't say anything about the existence of $\tau^*$.
> > >
> > > please see further discussion on pages 59 and 60 of [27] (Shiryaev).
> > >
> > > The stated condition when applied  to **finite-horizon** optimal stopping problems (the case considered here) is sufficient to guarantee existence of $\tau^*$,  since the optimal stopping time that achieves the above supremum **can be constructed explicitly** by the process of backward induction (See page 59, equation 3.7 for the construction process and Theorem 1 of Shiryaev [27] on  pg 60, equation 3.8 that shows that the construction achieves optimality).

---

> ### Comment · Senior_Area_Chairs · 2023-08-20
> **Please take a look at author response and let us know if your opinion has changed.**
>
> Thank you.

---

### Official Review · Reviewer_njDJ · 2023-07-06

**Soundness:** 3 good
**Presentation:** 3 good
**Contribution:** 3 good
**Rating:** 5
**Confidence:** 3

**Summary:**

Optimal stopping is the problem of choosing a time to take a given action based on sequentially observed random variables in order to maximize an expected payoff. Previous works used Deep Neural Networks to find the optimal stopping time (e.g. Backward Induction method), however, as the authors mentioned, these approaches have several limitations in non-Markovian settings. The paper presents the Optimal Stopping Problem as an inference problem on a Bayesian Network and they use RNNs to learn the model-free optimal stopping strategies.

**Strengths:**

- Authors introduce a reasonable way to solve the optimal stopping problem and provide the corresponding theoretical justifications.
- The approach is well motivated (lines 43-60).
- The experimental results demonstrate that their method outperforms the baseline.
- The authors compared the training and inference times for DROS and the baseline.
- The paper is well written and organized.
- Experiments were done on real-world benchmarks.


**Weaknesses:**

- There is a whole literature on how to use deep learning to solve partial differential equations (PDEs) in general, and more specifically option pricing problems. Many papers solve PDEs using deep learning in the context of optimal stopping time.
- There are several papers on optimal stopping problems, I encourage the authors to include more baselines in their comparisons.

**Questions:**

- American options are the most popular in practice and is a very good case study to solve.
- Have you tested your algorithm on different American options? If not, why not?


**Limitations:**

- From the tables 2 and 3 (Appendix C), the training time of DROS is bigger compared to the baselines.
- The proposed method can suffer in terms of time complexity when we test it to solve problems in high dimensions.

---

> ### Author Rebuttal · Authors · 2023-08-09
>
> We thank the reviewer for appreciating the motivation, contribution, presentation, and organization of the paper. The main concern seems to be with regard to the vast body of work on PDE based optimal stopping approaches, specifically with regard to solving American options.
>
> We hope that have fully addressed this concern in the specific comments below, including direct comparison of our model-free method with state-of-the-art PDE based American option solvers which are in fact model-based.
>
> Considering the mitigation of this perceived weakness, we hope the rating will also be reconsidered.
>
> > There is a whole literature on how to use deep learning to solve partial differential equations (PDEs) in general, and more specifically option pricing problems. Many papers solve PDEs using deep learning in the context of optimal stopping time. I encourage the authors to include more baselines in their comparisons.
>
> **Response:** Indeed, there is a lot of interest in using deep-learning to solve PDEs with application to option pricing. However, feel that PDE based baselines are not appropriate baselines for this paper, for the following reasons:
>
> * **The setting of the paper is discrete-time, finite-horizon optimal stopping problems**. There is no natural PDE which describe the dynamics such systems in general, thus ruling out PDE approaches. PDE based methods, including deep learning methods are inherently designed for *continuous-time* optimal stopping problems.  Note however, that while we may approximate continuous-time problems by choosing a fine discretization grid, to our knowledge, natively discrete-time problems cannot necessarily be solved using the PDE approach.
>
> * **We consider the PDE methods to be in the category of model-based methods**, since they start with a specific PDE to be solved. Consider, for example, popular PDE based American option pricing methods such as the Deep Galerkin Method (DGM) [Sirignano and Spiliopoulos, 2018] and the Backward Stochastic Differential Equation (BSDE) method [Chen and Wan, 2020]: These assume Markovian Black-Scholes Dynamics and the PDEs to be solved require the Black-Scholes model parameters, such as covariance of the Brownian motion, volatility, risk-free interest rate, and dividend yield. In contrast, our method (and those we compare against) does not use any prior information on the evolution dynamics of the underlying stochastic process.
>
> We will add references and discussion to reflect these points.
>
> > American options are the most popular in practice and is a very good case study to solve. Have you tested your algorithm on different American options? If not, why not?
>
> **Response:**   As stated above, this paper is about solving discrete-time, finite-horizon optimal stopping problems. Option pricing (specifically Bermudan options, that have discrete exercise opportunities) are just one example application. We did not include American options since they are continuous-time and hence we felt that they were not natural candidates to show efficacy of the developed methods.
>
> That said, our method can indeed be used to price American options in a model-free manner, simply by solving the corresponding Bermudan option at a finer discretization (ie: increasing discrete exercise opportunities to day level, for example). Note that Continuous-time methods for American options (such as those cited above) require discretization of the original PDE (ex: using Euler-Maruyama scheme) or random sampling (as used in DGM), so often do not end-up directly solving a Bermudan option.
>
> Although as noted above the PDE approaches are not model-free, **we have now run our approach in pricing challenging high-dimensional American options** and compare vs. published results from state of the art PDE baselines including Deep Galerkin Method (DGM) [Sirignano and Spiliopoulos, 2018] and the Backward Stochastic Differential Equation (BSDE) method [Chen and Wan, 2020].
>
> We consider the 100 dimensional continuous-time **American** geometric-average call option with Black-Scholes dynamics considered in [Sirignano and Spiliopoulos, 2018] and [Chen and Wan, 2020]. The option is characterized by the following parameters:
> $r$: 0.0, $\delta$: 0.02, $\sigma$: 0.25,  $\rho_{ij}$: 0.75, 'time_horizon_yrs': 2 years, 'strike_price': 100. The exact price of this option can be determined semi-analytically for comparison [Chen and Wan, 2020].
>
> | Method    | stock price | option_price | exact_price |
> | :---: | :---: | :---: | :---: |
> | [Sirignano and Spiliopoulos, 2018] | 100 | **9.9236** | 9.9345 |
> | [Sirignano and Spiliopoulos, 2018] | 110 | N/A | 15.6491 |
> | [Chen and Wan, 2020] | 100| 9.9187 | 9.9345 |
> | [Chen and Wan, 2020] | 110| 15.6219 | 15.6491 |
> | DROS-OSPG (ours) | 100| 9.8675 | 9.9345 |
> | DROS-OSPG (ours) | 110| **15.6428** | 15.6491 |
>
>
> Our model-free algorithm yields results competitive with state of the art PDE methods that assume Black-Scholes dynamics. We were surprised by the excellent performance of our method, since we did not expect it to work well in a natively continuous-time setting. This opens the door for using our approach to price American options in a model-free setting, especially when the underlying trajectories are non-Markovian and are not required to follow Black-Scholes dynamics.
>
> We can include a more comprehensive comparison with PDE methods and American options in the supplemental material.
>
> **References**
>
> [Sirignano and Spiliopoulos, 2018] DGM: A deep learning algorithm for solving partial differential equations. Journal of Computational Physics, 375:1339–1364, 2018.
>
> [Chen and Wan, 2020] Yangang Chen & Justin W. L. Wan (2020): Deep neural network framework
> based on backward stochastic differential equations for pricing and hedging American options in high dimensions, Quantitative Finance

---

> > ### Author Response · Authors · 2023-08-20
> >
> > Thank you for your valuable comments. We hope we have addressed all your concerns with regard to comparisons with PDE based optimal stopping methods and application to American option pricing. Since the deadline for author-reviewer discussion is fast approaching, please do let us know if there are any further clarifications.
> >
> > We would also like to highlight the several contributions (summarized in global comments to reviewers) to the under-researched area of **optimal stopping in non-Markovian settings**, which has several real-world applications, including computational finance. This is achieved by bringing together, for the first time, RNNs, probabilistic graphical models and policy-gradient methods. One of the key contributions is a new policy-gradient algorithm for optimal stopping that avoids expensive Monte-Carlo rollouts by performing inference on a Bayes Net model of state-action trajectories. This opens the door to new applications.  For instance, in computational finance, asset dynamics are often modelled with Markovian Black-Scholes type models.  In many real-world settings, such assumptions are invalid. Our approach (RNNs and optimal stopping policy gradient methods) provide an elegant alternative, especially in non-Markovian settings.
> >
> > The results of this paper would be of significant interest to the NeuRIPs community. We hope this provides sufficient grounds to reconsider the rating of the paper.

---

> > ### Comment · Reviewer_njDJ · 2023-08-22
> > **Thank you for your response**
> >
> > I have read all the comments. I want to thank the authors for running the new experiments regarding the American Options. I would be happy to increase the score if the authors have the possibility to run more experiments with several strike prices in the case of American pricing so we can draw strong conclusions. Can you please update the code and include the American option benchmarks?

---

### Official Review · Reviewer_XsN5 · 2023-07-09

**Soundness:** 3 good
**Presentation:** 2 fair
**Contribution:** 3 good
**Rating:** 5
**Confidence:** 2

**Summary:**

The paper proposes an RNN-based approach for optimal stopping which is based on a Bayesian inference view of optimal stopping. The proposed model can be trained with direct optimization via policy gradients, or with expectation-maximization (EM). These two appraoches are shown to be equivalent. This new RNN-based approach is shown to outperform state of the art methods on a couple of commonly used datasets.

**Strengths:**

The limitations of the existing deep neural network approaches are discussed and contributions of this work are clearly stated. The background on optimal stopping is also introduced in detail.

**Weaknesses:**

- **Writing and organization of the paper have much room for improvement.**

**Overuse of abbreviations makes the paper difficult to read.** Although abbreviations like DNN and EM are common and inevitable, I personally would suggest against abbreviating weighted maximum likelihood as WML and policy gradiants as PG in the text. To make things worse, one of the main approaches is named DROS-OSPG, with an unnecessary repeat of "optimal stopping" which I find to be cumbersome and confusing.

**Equations with conflicting real and dummy indices.** In equation 2 and theorem 3.1, the index $j$ is both used as a real index and a dummy index for summation.

**There's too much technical detail in the main paper.** Equivalence of EM and policy gradient is interesting but probably belong better in the supplements. However I do understand that the significance of this fact might have eluded me since I am not an expert in this field. The Keras-specific implementation detail on lines 246-247 can also be saved for the supplements.

**No conclusion or discussion paragraph at the end.** As a consequence, the paper does not find room to discuss its limitations, which is a requirement for NeurIPS papers.

- **Ablations are missing.** Since I am not an expert in this specific field, I do not know if the improvements over the state of the art are significant enough. From the prospective of a research paper, I feel like some design decisions need to be justified by running ablations. For example, how important are the weights in the weighted maximum likelihood objective? Abalation studies like this would justify the various claims put forth throughout the paper.
---
Once again I must say that it is very likely that I do not understand the significance of the paper due to lack of familiarity with the field. My main concerns with the paper lie in its presentation. I do not think the paper at its current state is doing a good job of illustrating the key ideas of the new method and of convincing me that the contributions are novel and significant.

**Questions:**

- In the first equation in equation 2, why is the numerator $R_\tau$?
- What is the justification for the paramterization of $Y_j$ in equation 2? Is this an arbitrary design decision?
- Why is the XOR used for $Y$ on line 167?
- In what sense is the proposed method a Bayesian interpretation of optimal stopping? Does the Bayesian interpretation still hold once the weights are introduced into the objective?

**Limitations:**

The authors did not adequately address the limitations of the work since there is no conclusion or discussion section.

---

> ### Author Rebuttal · Authors · 2023-08-09
>
> We appreciate the reviewers comments. To better help clarify the novelty and significance of the paper and approach, we have included context in the global comments to all reviewers. We hope this helps the reviewer in appreciating the contribution and to reconsider the rating.
>
> Specific questions raised in the review are addressed below.
>
> >In the first equation in equation 2, why is the numerator $R_{\tau}$?
>
> **Response:** This is a typo:
>
> $\mathbb{P}(Y_j = 1 | {\bf R}_H, A_j = 1)$ should be equal to
>
> $\displaystyle\frac{R_j}{\sum_{k=0}^H R_k }$
>
> instead of
>
> $\displaystyle\frac{R_{\tau}}{\sum_{j=0}^H R_j}$
>
> as appeared in the paper.
>
> >What is the justification for the parameterization of $Y_j$ in equation 2? Is this an arbitrary design decision?
>
> **Response:** The formulation of this conditional probability distribution is indeed a design choice, but not arbitrary.
>
> * Since we desire to encode reward opportunities at each step in the trajectory into the BN and every node in the BN represents a conditional probability distribution (CPD), it is natural to use the relative reward at a time-step (normalized by total reward over the trajectory) to represent this reward opportunity as a CPD.
>
> * A key consequence of this choice is that it leads to an equivalence with policy gradients and minimization of the optimal stopping objective.
>
> Similar design choices have also been made in other related work. Please see equation (3) in [15] Sergey Levine, "Reinforcement Learning and Control as Probabilistic Inference: Tutorial and Review". The authors say: "While this might at first seem like a peculiar and arbitrary choice, it leads to a very natural posterior distribution..."
>
> Note that while we are inspired by a corresponding view of reinforcement learning [15], the resulting modeling choices needed to capture the structure of optimal stopping in our approach leads to a very different BN model. For instance in the former case, rewards may be accumulated over time-steps and trajectory lengths are not variable, rolling out to the horizon. Also that formulation leads to a maximum entropy objective.
>
> > Why is the XOR used for on line 167?
>
> **Response:** We use the XOR operator to define the random variable $Y$ from the $Y_j$'s. Since $Y=1$ if and only if exactly one of the $Y_j$'s are 1. The reward for a trajectory can only be claimed by a stop action at a single time-step. This requires only a single $Y_j = 1$, allowing us to sum the probabilities of stopping and collecting rewards at each time-step of the trajectory (equation 5)
>
> >Ablation for the weights in WML
>
> **Response:** Our intent in this paper is not to use the WML weights explicitly. We provide a particular (*and natural*) choice of weights: **weight each trajectory by the expected reward of that trajectory** which results in a particular WML objective, which then leads to equivalence between the WML and policy-gradient approaches **and** minimization of the original optimal stopping objective from definition 2.4.
>
> Also, we feel that it is not meaningful to ablate different weight choices in this case since the WML objective itself changes with different weights. To ablate, one needs to change something and measure against a fixed target. In this case the proposed ablation would be shooting at a different target.
>
> >In what sense is the proposed method a Bayesian interpretation of optimal stopping?
>
> **Response:**  What we have shown is that the peculiar state-action trajectories in optimal stopping can be modelled explicitly by a Bayesian Network (Figure 1a) which can then be used to rewrite the classic optimal stopping objective of equation (10) in the form of equation (12). It is in this sense that we have a Bayesian Net interpretation of the optimal stopping problem. A real benefit of this view is that by computing probabilities over state-action trajectories, we avoid explicitly sampling actions (a.k.a. Monte-Carlo state-action trajectory rollouts) like policy-gradient methods typically do.
>
> A reward augmented version of the state-action trajectory network (Figure 1b) additionally captures the notion of an optimal stopping trajectory, by introducing optimality variables $Y_j$ and corresponding conditional probability distributions (equation 2) into the model. This latter approach leads to a WML problem which does not in general reduce to minimization of the classic optimal stopping objective, **unless** the CPD and weights have the specified form. In this sense, this is a generalized interpretation of optimal stopping problems.
>
> >Does the Bayesian interpretation still hold once the weights are introduced into the objective?
>
> **Response:**  The reward augmented Bayes Net model (Figure 1b) is simply a model of state-action trajectories and corresponding relative reward possibilities **inside** a  trajectory. This **interpretation does not change** by introducing weights into the objective since these weights assign importance to entire trajectories (so do not affect relative rewards inside a trajectory).

---

> > ### Comment · Reviewer_XsN5 · 2023-08-16
> >
> > I thank the authors for their detailed response. I think the response did help me better understand the work and its significance. Given that the methodology and results presented in the paper all seem solid and the typos are fixed, I am willing to bring my score up to a 5. Thanks!

---

> > > ### Author Response · Authors · 2023-08-19
> > >
> > > We thank you for your valuable comments and kind reconsideration of our paper.

---

### Author Rebuttal · Authors · 2023-08-09

We thank the reviewers and appreciate the concerns raised. Here we address concerns regarding the significance of our contribution. We respond to each reviewer individually in specific comments.

While there is a vast body of work on optimal stopping problems in the Markovian setting, the literature on model-free optimal stopping in **non-Markovian settings** is sparse. This is of great practical importance in areas such as finance (option pricing), operations research (predictive maintenance), early classification/detection etc.

Two of the main challenges (mentioned in lines 43-60, 76-79) in developing effective algorithms for this setting include:

* **Explosion of state space:**.  Ready extensions of Markovian approaches to non-Markovian settings results in state space explosion rendering them impractical. Solving the problem at hand **requires efficient parameterization of state space, such as afforded by RNNs**. However, popular optimal stopping approaches either cannot use RNNs for structural reasons (backward induction) or fare poorly in non-Markovian settings even if RNNs are used (fitted Q-iteration). Thus, *one does not encounter the use of RNNs in optimal stopping settings*.

* **Lack of model-free direct RL methods:**  RL style policy-gradient algorithms are typically online algorithms and require **expensive monte carlo policy rollouts**. *Policy gradient methods are notably missing* from the optimal stopping literature.

Keeping these problems in mind, **we bring together, for the first time, RNNs, probabilistic graphical models and policy-gradient methods to design an RNN-based policy-gradient algorithm for non-Markovian optimal stopping settings**. One of the key contributions is to is to *avoid expensive Monte-Carlo rollouts in the policy gradient algorithm by performing inference* on a Bayes Net (a probabilistic graphical model) model of state-action trajectories (Section 4).

Extending the Bayes net trajectory model with reward augmentation (section 3) yields a weighted maximum likelihood (WML) policy estimation approach that is a *generalization of the optimal-stopping policy-gradient method, in the sense that we recover our policy gradient algorithm for specific design choices of CPDs and weights* (section 4.1). One of the key benefits of this generalization is that the procedure can be adapted to various settings by augmentation with additional variables/latents makes it possible to model time-dependent stochastic disorders, such as change-points etc, also permitting a readymade solution method, via incremental EM.

---

### Decision · Program_Chairs · 2023-09-21

**Decision:**

Accept (poster)

**Comment:**

The reviewers have provided a comprehensive evaluation of the paper, which proposes an RNN-based approach for optimal stopping based on a Bayesian inference view. The reviewers have highlighted the strengths of the paper, including the clear statement of contributions, detailed introduction of optimal stopping, and the theoretical justifications provided for the proposed approach. The experimental results demonstrating the method's superiority over the baseline were also appreciated.

However, the reviewers have also pointed out several areas for improvement. The presentation of the paper, particularly the use of abbreviations and the organization of technical details, was a common concern. The lack of a conclusion or discussion section was also noted, as was the absence of ablation studies to justify design decisions.

The reviewers also raised several questions about the paper's content, including the justification for certain parameterizations and the Bayesian interpretation of optimal stopping. These questions indicate areas where the paper could provide more clarity or justification.

Despite these concerns, the reviewers generally found the paper to be technically solid and its contributions to be valuable. The paper's approach to optimal stopping is novel and its performance on commonly used datasets is promising.

In light of these evaluations, I recommend accepting this paper. However, I strongly encourage the authors to consider the reviewers' feedback in revising their paper. Improving the presentation and organization, providing a conclusion or discussion section, and addressing the questions raised by the reviewers could significantly enhance the paper's clarity and impact.